# Spectro-temporal encoded multiphoton microscopy and fluorescence lifetime imaging at kilohertz frame-rates

Sebastian Karpf [1,2 ✉], Carson T. Riche[3], Dino Di Carlo[3], Anubhuti Goel[4], William A. Zeiger[4], Anand Suresh[4], Carlos Portera-Cailliau [4] & Bahram Jalali[1,3]

Two-Photon Microscopy has become an invaluable tool for biological and medical research, providing high sensitivity, molecular specificity, inherent three-dimensional sub-cellular resolution and deep tissue penetration. In terms of imaging speeds, however, mechanical scanners still limit the acquisition rates to typically 10–100 frames per second. Here we present a high-speed non-linear microscope achieving kilohertz frame rates by employing pulse-modulated, rapidly wavelength-swept lasers and inertia-free beam steering through angular dispersion. In combination with a high bandwidth, single-photon sensitive detector, this enables recording of fluorescent lifetimes at speeds of 88 million pixels per second. We show high resolution, multi-modal - two-photon fluorescence and fluorescence lifetime (FLIM) – microscopy and imaging flow cytometry with a digitally reconfigurable laser, imaging system and data acquisition system. These high speeds should enable high-speed and high-throughput image-assisted cell sorting.

[1] Department of Electrical Engineering and Computational Science, University of California, Los Angeles (UCLA), Los Angeles CA-90095, USA. [2] Institute of Biomedical Optics (BMO), University of Luebeck, 23562 Luebeck, Germany. [3] Department of Bioengineering, University of California, Los Angeles (UCLA), Los Angeles CA-90095, USA. [4] Department of Neurology, University of California, Los Angeles (UCLA), Los Angeles CA-90095, USA. ✉email: sebastian.karpf@uni-luebeck.de

The extension of regular one-photon fluorescence microscopy to non-linear two-photon microscopy (TPM)[1] has led to important applications, e.g., in brain research, where the advantages of deeper tissue penetration and inherent three-dimensional sectioning capability enable recording neuronal activity to interrogate brain function in living mice[2]. Further, fluorescence lifetime imaging (FLIM)[3] can probe internal biochemical interactions and external environment of molecules, e.g., for quantifying cellular energy metabolism in living cells[4,5]. The quadratic dependency on intensity of TPM favours laser-scanning over whole-field illumination, so typically raster-scanning with galvanometric mirrors is conducted. However, these mechanical mirrors are inertia limited to line-scan rates of <20 kHz. This limits the imaging frame rate, however faster frame-rates are desirable, e.g., for neuronal activity imaging at >1000 Hz frame-rates[6–8]. Therefore, new technologies were developed with faster acousto-optical scanners[6,9], polygonial scanners[10], parallelized multi-foci excitation[11–13], optical scanning[8] or sparse sampling[7], just to name a few. For FLIM, recent developments also increased imaging speeds[14] with techniques employing multifoci[15] or widefield imaging[16–19] or increased detection speed by analogue lifetime detection[20–22] which permitted speeds up to video-rate[23]. Enhancing imaging speeds through spectral-encoded scanning has been successfully employed for confocal microscopy[24], high-speed brightfield imaging[25], quantitative phase imaging[26] and more, especially by employing the time stretch technique[25–27]. However, these fast imaging approaches could not be used for fluorescence imaging, as both the emission spectrum and the fluorescent lifetime are independent on the excitation wavelength (Kasha's rule), so the original spectral encoding is lost upon the fluorescence emission. We present a solution to this problem by further employing temporal encoding from a wavelength swept laser. This concept achieves spectro-temporal encoded imaging, where the wavelength is used for high-speed inertia-free point scanning and the temporal encoding for one-to-one mapping of the signal to the imaging pixels. This temporal encoding is just like in conventional raster-scanning or laser scanning microscopy. Spectro-temporal encoded imaging has unique advantages over other high-speed non-linear imaging approaches[6–9,11–13] in terms of resolution, lifetime modality, compactness, flexibility, and fibre-based setup.

Here we report on a high-speed laser scanning technique for non-linear imaging using a rapidly wavelength swept laser in combination with a diffraction grating to achieve inertia-free, very rapid beam-scanning, orders of magnitude faster than mechanical scanners. We employ a high-speed swept source Fourier-domain Mode-locked (FDML) laser[28] which is modulated to short pulses and amplified to high-peak powers. This FDML-MOPA laser was previously described in detail[29]. The swept wavelength output is sent onto a diffraction grating for line scanning (Fig. 1). Each pulse illuminates a distinct pixel both in time and in space (spectro-temporal encoding). The y-axis is scanned with a galvanometric mirror at typically 1 kHz speed (slow axis). At an FDML sweep rate of 342 kHz this achieves a frame rate of 2 kHz for a 256 × 170 pixel frame size and 88 MHz pulse repetition rate. A high-bandwidth detection at single-photon sensitivity enables recording of second harmonic generation (SHG), Two-Photon fluorescence and fluorescent lifetime imaging (FLIM) at speeds up to the excitation rate of 88 million pixels per second. This fast two-photon microscope, which we coin spectro-temporal laser imaging by diffracted excitation (SLIDE) microscope, can be digitally programmed to allow for adapting the pulse repetition rate to the fluorescent lifetimes. This is achieved through the direct pulse modulation through a 20 GHz bandwidth electro-optic modulator (EOM) (cf. Fig. 1 and our previous report[29]).

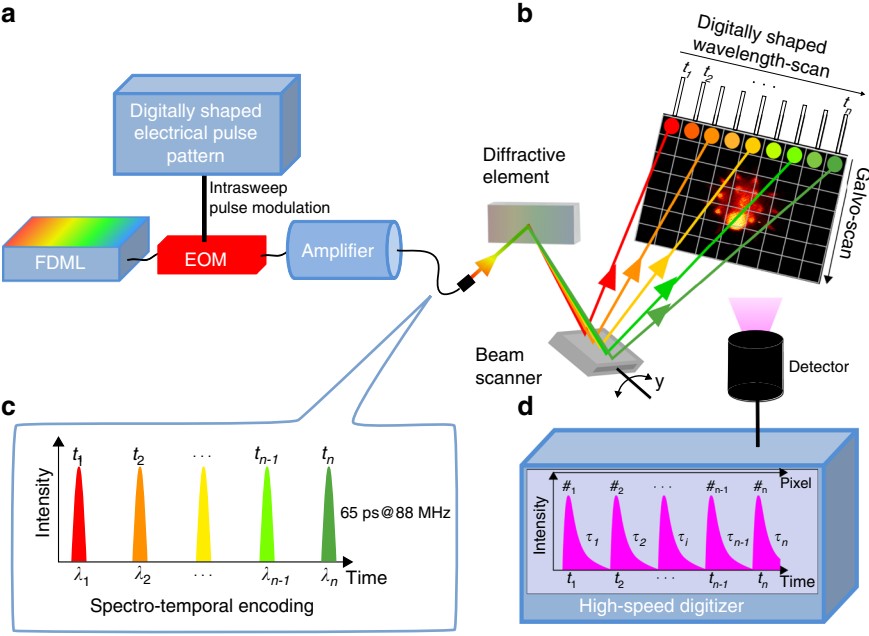

**Fig. 1 Principle of spectro-temporal laser Imaging by diffractive excitation (SLIDE).** **a** A swept-source Fourier Domain Mode-Locked (FDML) laser is pulse-modulated, amplified and diffracted to produce rapid beam steering through spectrum-to-line mapping (**b**). The mapping pattern along each line is digitally shaped through the pulse modulation onto a fast electro-optic modulator (EOM). For raster-scanning the y-axis is scanned using a galvanometric scanner. **c** In SLIDE, each pulse has both a unique wavelength and time (spectro-temporal encoding) leading to a sequential and pixelwise illumination. **d** Fluorescence signals are recorded at high-detection bandwidth to also enable fluorescent lifetime imaging (FLIM) at high speed.

## Results

**Picosecond excitation pulses for rapid lifetime imaging.** The fast lifetime imaging capability is possible by the direct analogue recording of the fluorescent lifetime decay and is further enhanced by the higher number of photons generated per pulse by picosecond excitation pulses[30,31], enabling single pulse per pixel illumination[21]. This has a number of advantages over traditional illumination: (i) A single pulse per pixel leads to a very low effective repetition rate per pixel, equal to half of the framerate (~1 kHz) for bi-directional scanning. This has been shown to decrease photobleaching and thereby increasing signal levels[32]. As a consequence, even when averaging is desired, e.g., for double-exponential lifetime fits, it may be advantageous to average 2000 frames obtained with SLIDE within 1 s, as opposed to pixel averaging by applying 2000 pulses per pixel with a pixel dwell time of 23 μs and then raster-scanning the image in 1 s. (ii) Longer pulses lead to reduced pulse peak powers at same SNR, thus having the advantage of avoiding higher-than-quadratic effects like photobleaching[32,33] and photodamage[34–36] (scale at orders >2). Thus, the sample can be imaged for comparatively longer times. Still, the picosecond pulses are shorter than timescales for intersystem crossing (ISC), so no further excitation from the triplet state should occur within the same pulse. (iii) The

longer pulses are generated by digitally synthesized EO modulation, which renders the excitation pattern freely programmable. For example, for optimal detection the pixel rate can be tailored to the fluorescence lifetime of the sample and allows warped (anamorphic) spatial illumination that takes advantage of sparsity to achieve optical data compression[37]. (iv) Longer pulses generate quasi-monochromatic light and this renders the high-speed line-scanning spectral mapping by diffraction gratings possible. (v) The quasi-monochromatic light is optimally compatible with fibre delivery by omitting chromatic dispersion and pulse spreading. The excitation laser presented here is already fully fibre-based, making it compatible with a future implementation into a multiphoton endoscope.

**Time bandwidth product in spectro-temporal laser imaging by diffractive excitation (SLIDE).** Spectro-temporal encoding with high-information density places rigorous requirements on the spectro-temporal bandwidth of the light source[37]. The wavelength sweep time $\Delta T$ is equal to the number of pixels n times the time between pulses $\Delta t_i$, governed by the information-interrogation induced latency (here: fluorescent decay time $\tau_i$). Considering, for example, 256 horizontal (linescan) pixels and a

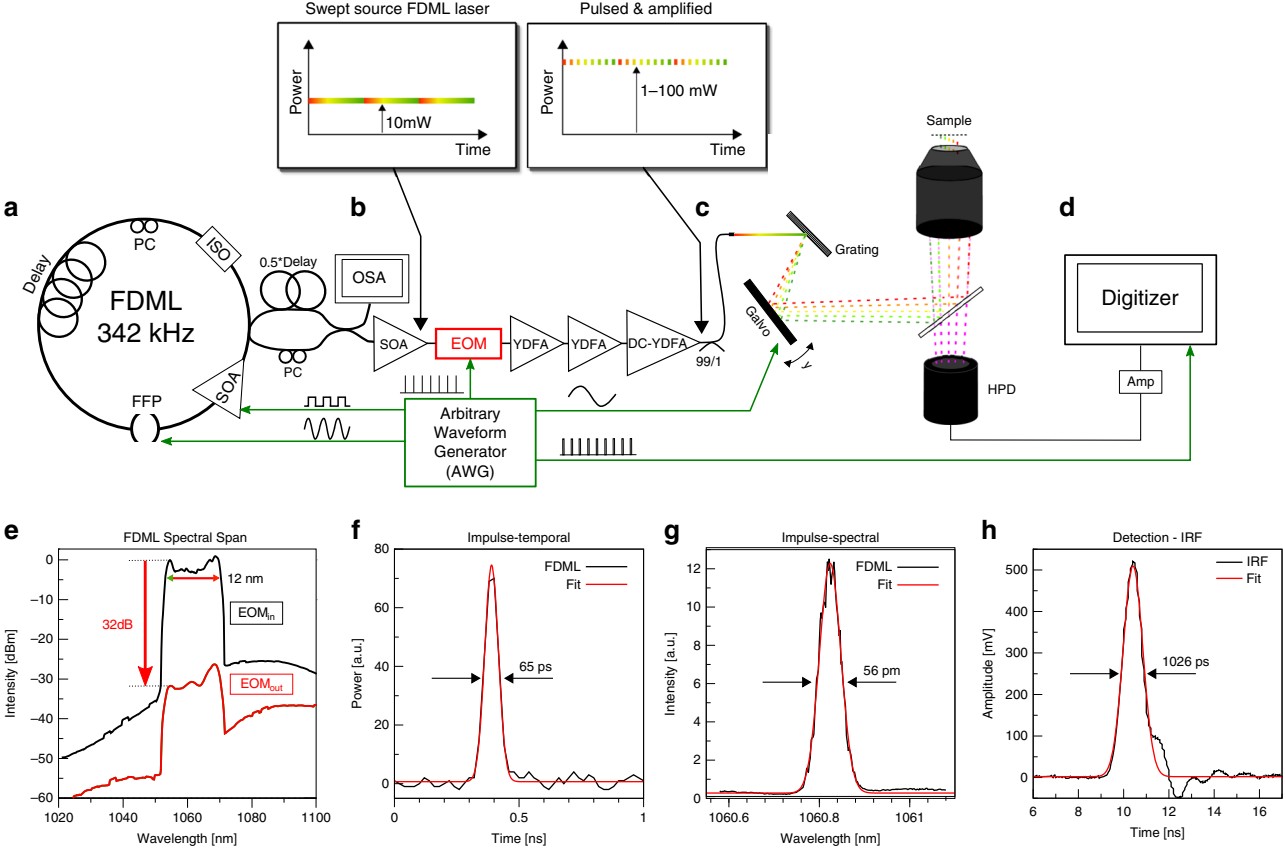

**Fig. 2 Experimental Setup of the SLIDE system. a** The light source is an FDML-MOPA laser[28,29] at 1060 nm (±6 nm) and 342 kHz sweep repetition rate. The FDML-MOPA laser was already described in detail in a previous report[29]. **c** The wavelength-swept laser is modulated by a digitally programmed pulse pattern which determines the line mapping in SLIDE. Typically, each sweep (2.9 μs duration) is modulated to 256 impulses of short temporal width (65 ps) to achieve a pixel rate of 88 MHz. This leaves enough time for most fluorescence transients to decay. The whole system is synchronized through an arbitrary waveform generator (AWG) which drives the FDML-MOPA laser[29], the y-axis galvo and the digitizer card (trigger and sample clock). A high-numerical aperture (NA) objective focuses the excitation light on the sample and collects the epi-generated fluorescence signal. A dichroic filter directs only non-linear signals on a fast hybrid photodetector (HPD), connected to a transimpedance amplifier and a fast digitizer (**d**). **e** A wavelength span of 12 nm is used for spectral scanning. The EOM modulator achieves a high extinction ratio[29] (~30 dB optical). **f** The recorded temporal pulse width achieved with the EOM is 65 ps. **g** High-resolution pixel mapping is possible due to the narrow instantaneous linewidth of the pulsed FDML laser pulses measured to 56 pm. **h** The instrument response function (IRF) of the detection was determined by SHG in a urea sample to 1026 ps FWHM.

typical total fluorescent decay time of 10 ns, this time calculates to $\Delta T = 2.56$ µs. Assuming a spectral resolution of $\Delta\lambda_i = 100$ pm for the diffractive mapping means that the light source needs to sweep over $\Delta\lambda = 25.6$ nm in $\Delta T = 2.56$ µs. A unique feature in SLIDE is that the spectro-temporal bandwidth scales quadratically with the number of pixels (in linescan):

$$\text{Spectro} - \text{temporal bandwidth } M_{ST} = \Delta T \times \Delta\lambda = n^2 \times \Delta\lambda_i \times \Delta t_i \tag{1}$$

A wavelength tuning speed of tens of nm over few microseconds is beyond the reach of conventional tuneable lasers[28,38]. Although very fast tuning speeds can be achieved by chirping a supercontinuum pulse source in a dispersive medium as employed in time stretch techniques, achieving a time span of 2.56 µs is about three orders of magnitude beyond the reach of available dispersive elements (typically in the ns-regime). Furthermore, the spreading of energy due to the stretching would result in negligible peak powers and would prevent non-linear excitation.

Spectro-temporal stretch via an FDML laser solves this predicament[37]. The FDML laser provides a combination of large spectral span[39], along with microseconds time span and narrow instantaneous linewidth. This type of laser has mainly been used for fast optical coherence tomography (OCT)[28,38,40], semiconductor compressed pulse generation[41] and non-linear stimulated Raman microscopy[42] and recently inertia-free LiDAR[37]. Its low instantaneous linewidth allows us to achieve a high spatial resolution below 1 µm (see Supplementary Fig. 6), a feat that is not possible with chirped supercontinuum sources.

**SLIDE: Experimental setup**. The experimental setup of the SLIDE microscope is presented in Fig. 2. The whole system is electronically synchronized by an arbitrary waveform generator (AWG). It drives the FDML-MOPA laser including the pulse pattern, the $y$-axis galvo mirror and also the trigger and clock of the digitizer card. The modulated pulse length was measured to be 65 ps (Fig. 2g). This pulse width corresponds to a time-bandwidth calculated linewidth of 25 pm, so only a factor of two lower than the measured linewidth[29]. Raster scanning on the sample is achieved through spectro-temporal encoding along the $x$-axis and mechanical scanning on the $y$-axis (cf. Fig. 1b). It shall be noted that the used spectral bandwidth of 12 nm (Fig. 2e) lies well within most Two-Photon absorption bandwidths, so after absorption and excited state relaxation (Stokes shift), the fluorescence emission is equivalent for all pixels along the whole line (Kasha's rule). The $x$-axis imaging resolution is given by the spectral resolution of the grating (67 pm, see methods) and the instantaneous laser linewidth (56 pm, Fig. 2g). In combination with the swept bandwidth of 12 nm, this should allow for 12 nm/0.067 pm ≈ 180 discernible spots and thus a lateral resolution of 100 µm/180 = 556 nm for the 60x objective (see Methods). We moderately oversample the $x$-axis by programming 256 pulses along the $x$-direction. We tested the imaging resolution achieved through spectro-temporal encoding by imaging 100 nm fluorescent beads (Supplementary Fig. 6) which yielded a resolution of 596 nm in $x$-direction (spectral scanning direction) and 455 nm in $y$-direction for the 60× NA 1.4 objective, i.e., close to the calculated resolution and sufficient for high-resolution imaging of sub-cellular features. We further verified that the fluorescence signal scales quadratically with the illumination power confirming Two-Photon origin of the signal. The instrument response function (IRF) of the detection was determined by SHG from a Urea sample and fitted to 1026 ps (Fig. 2h). The validity of the analogue lifetime detection approach was already confirmed in a previous work[30], where we compared the recorded lifetime value of Rhodamine 6 G to a literature value in good agreement. This analogue detection enables rapid FLIM

microscopy at even only a single pulse per pixel illumination (SP-FLIM)[21], as discussed above.

**Imaging performance**. SLIDE imaging performance was assessed first by recording the field of view (FOV) on a resolution target (Fig. 3b). The $x$-axis was scanned with the spectro-temporal encoded scan while the $y$-axis was scanned with a galvanometric mirror (Fig. 3a). Figure 3c shows a two-photon microscopy image of a *euglena gracilis* microalgae and was recorded within 497 µs at 2 kHz frame rate. The time trace (Fig. 3d) of a single line illustrates the high SNR of up to 490 (peak SNR). This is due to the high photon counts achieved per excitation pulse (note that a single photon generates a voltage of 50 mV). By zooming in on two individual pixels, we see a difference in the transient fluorescence decay time (Fig. 3f). To this end, $3 \times 3$ spatial binning was performed to yield enough fluorescent photons for a mono-exponential fit[43]. By fitting with the convoluted IRF a FLIM image can be generated (Fig. 3e) revealing two different domains within the algae cells. The autofluorescent chloroplasts decay fast and are colour-coded red, while Nile Red, an exogenous fluorophore which was added to highlight lipid generation within the microalgae, has a longer decay time and is colour-coded green. Both imaging modalities TPM and FLIM are extracted from the same data which was acquired within 497 µs for this $256 \times 170$ pixel image. Further imaging examples can be found in the supplementary material (Supplementary Figs. 1, 2, 4–6).

**SLIDE imaging flow cytometry**. To calibrate the speed and accuracy of SLIDE we performed imaging flow cytometry of five different species of fluorescent beads (Fig. 4a, b). The beads range in size from 2 to 15 µm and are thus chosen as examples for typical mammalian cell sizes, although perhaps not comparable in terms of brightness. Figure 4 nicely showcases that SLIDE imaging flow cytometry is capable of obtaining high quality images even at these high throughput rates of >10,000 objects per second. Fig. $4d_1$–$d_4$ were acquired within 497 µs per image. Since these beads show high signal levels, analysis of the time domain data revealed that about 1–10 fluorescent photons were achieved per excitation pulse at 15 mW average excitation power. After $9 \times 9$ spatial binning was applied, photon counts were >100 photons per lifetime curve for reliable mono-exponential fitting[43]. For this application, tail-fitting was applied assuming lifetimes significantly longer than half of the IRF. This approach will achieve fast fitting and still yield reliable qualitative lifetime contrast in order to distinguish different lifetimes. As can be seen by exemplary lifetime fits for all five beads shown in Fig. 4c, the lifetime accuracy was still high, and the measured signals enabled high fidelity lifetime fits.

As a biological sample we imaged *Euglena gracilis* cells in SLIDE imaging flow cytometry mode, where autofluorescence from chloroplasts revealed a short fluorescence lifetime and Nile Red stained lipids show a longer lifetime (see also Fig. 3). Figure 5 presents four different snapshots of *Euglena gracilis* microalgae in flow where each image pair (Two-Photon & Two-Photon FLIM images) was acquired within 1 ms. The TPM images already show the high resolution which reveals sub-cellular morphology. Yet, through the FLIM images it is possible to discern chloroplasts from lipids through their fluorescent lifetime. The images have very high resolution and high SNR and, even though they were obtain at very high speeds in flow, they have quality comparable to regular one-photon fluorescence microscopy[44]. Excitation power was 30 mW on the sample and $5 \times 5$ pixel spatial binning was applied to ensure >100 photons per lifetime curve. Fluorescence lifetime can aid in discerning lipids as it is independent of concentration and signal height. This high-speed lipid content screening by SLIDE FLIM

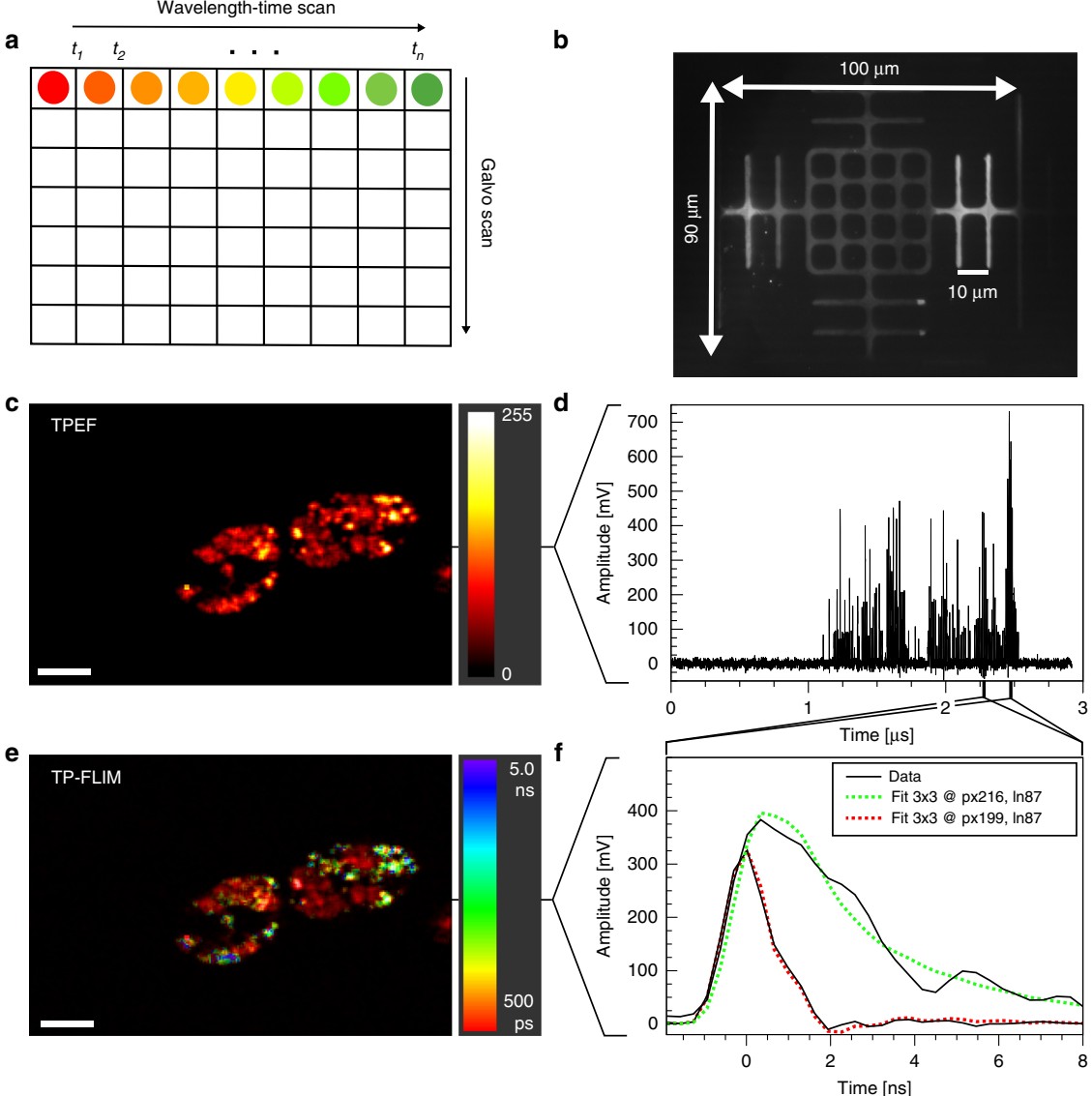

**Fig. 3 SLIDE high-speed dual-modailty imaging at 2000 frames per second. a** In SLIDE microscopy, the x-axis is scanned by spectro-temporal encoded diffraction scanning and the y-axis by a galvo mirror. **b** The field-of-view (FOV) is 100 × 90 μm² using the 60x objective (resolution target has 10 μm pitch). **c** Two-Photon intensity image of *Euglena gracilis* algae cells (256 × 170 pixels). Each line is acquired within 2.9 μs at high signal-to-noise ratio (SNR) of up to 490 (peak SNR) (cf. **d**). Zooming into individual pixels (**f**) reveals different transient fluorescent decay times which can be fitted to extract a fluorescence lifetime value for each pixel. Therefore, a 3 × 3 pixel binning was applied for higher lifetime fidelity. The lifetimes are colour-coded to achieve a TP-FLIM image (**e**) from the same dataset. In the TP-FLIM image the rapidly decaying autofluorescent chloroplasts (red) can be distinguished from the Nile Red-stained lipid droplets (green, blue). This unaveraged image was acquired at 2.012 kHz frame-rate (497 μs recording time). The average power on the sample was 30 mW. Lifetimes were determined by deconvolution with the IRF (convolution fit shown in **f**). Photon counts for the curves in (**f**) are 82 photons (red) and 246 photons (green). Scale bar represents 10 μm.

cytometry may help in purification of high lipid content microalgae for efficient biofuel production[44].

## Discussion
It shall be noted that the high speeds achieved by SLIDE require bright samples in order to detect many fluorescence photons per excitation pulse. In fact, the high speed of 88 MHz pixel rate was achieved for two-photon fluorescence imaging, yet for FLIM imaging pixel binning was applied to achieve the necessary photon numbers of >100 photons per pixel[43]. This blurs the lifetime (colour) resolution, without compromising the morphological resolution, which originates from the fluorescence intensity images. However, not all samples provide sufficient

fluorophores in the focal volume, such as genetically encoded fluorescent proteins. We tested imaging of genetically encoded tdTomato fluorescent proteins in ex vivo mouse brain to test whether a kilohertz frame-rate could be achieved in imaging neuronal activity. However, even though we were able to obtain morphological images at 1 kHz rate (see Supplementary Figs. 1, 2, 4), we found that only single fluorescent proteins were present in the focal volume and the signal was saturated, as increasing the laser power did not yield a quadratic signal increase. This does not permit recording neuronal ensemble activity at the kHz speed of the SLIDE systems, as both fluorescence intensity changes or fluorescence lifetime changes require tens to hundreds of photons thus sacrificing the speed as a result of averaging. Therefore, biochemically engineered fast and bright fluorescent proteins are

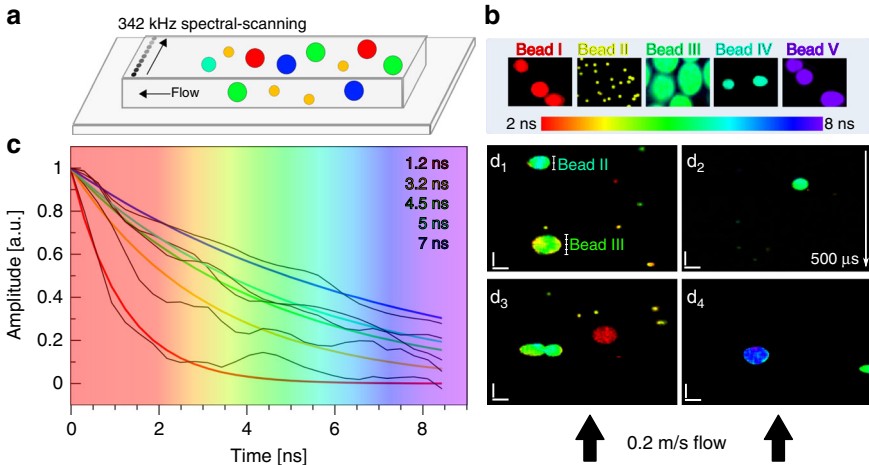

**Fig. 4 SLIDE Imaging flow cytometry at 0.2 m/s flow speed. a** SLIDE line imaging was performed at 342 kHz line-scanning rate inside a flow channel. Only the direction perpendicular to the flow was scanned via SLIDE, while the flow speed gives the scanning in the *y*-axis direction. **b** Five different calibration fluorescent beads were first imaged at rest to obtain reference lifetime colours. Subfigure **b** shows averaged SLIDE images (1000-times averaging). The lifetimes range from 2–8 ns (**b**, **c**). Afterwards, a mixture of all beads was imaged in flow at flow speeds of 0.2 m/s in order to obtain even sampling period in both dimensions at 550 nm pitch (see Methods). The five different species show different lifetimes (**c**) and can be clearly distinguished via TP-FLIM ($d_1$–$d_4$) even at these high flow speeds. Sample throughput is estimated to be >10,000 beads per second. Images were generated with 170 vertical lines, so each image $d_1$–$d_4$ was recorded within 497 μs. The images have high SNR and high resolution. For lifetime accuracy, 9 × 9 pixel binning was applied to yield >100 photons per fluorescence lifetime decay to allow for reliable mono-exponential lifetime fits[43]. The pixel rate was 88 MHz, i.e., single excitation pulse per pixel without averaging. The power used was 15 mW, scale bars represent 10 μm.

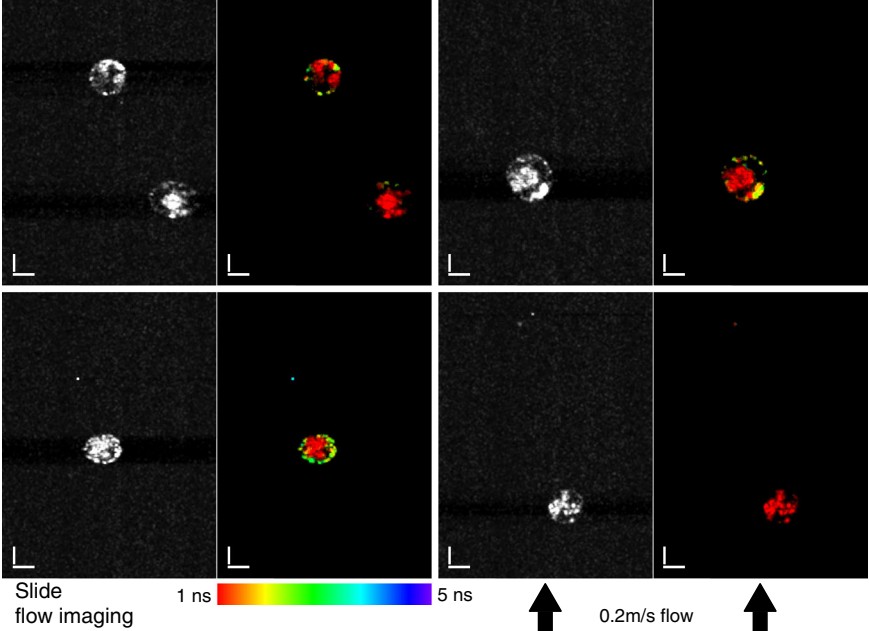

**Fig. 5 SLIDE Lifetime Imaging flow cytometry of *Euglena gracilis* algae cells at 0.2 m/s flow speed.** The Nile Red stained lipids can be clearly distinguished from the autofluorescent chloroplasts by their longer fluorescent lifetime (~5 ns and ~1 ns, respectively, cf. Fig. 3), even at these high flow rates. High subcellular resolution is achieved and morphologic structure as well as molecular information is recorded by SLIDE imaging flow cytometry. The displayed images have 256 × 342 pixels resolution which corresponds to a recording time of only 1 ms per image pair (TPM left, SP-FLIM[21] right). In the FLIM images a 5 × 5 spatial binning was applied to ensure >100 photons per fluorescence lifetime decay. The power used was 30 mW, scale bars represent 10 μm.

needed, which can also be expressed at high abundance without interfering with cellular behaviour, in order to enable kHz frame-rate imaging of neuronal activity for SLIDE or any other fast technique in the future.

It is noted that SLIDE microscopy employs higher average powers (~10–100 mW) compared to conventional two-photon microscopes (1–30 mW). However, we did not notice any sample damage. The higher powers are due to the fast acquisition rates obtained by SLIDE together with the relatively longer pulse length. It shall be noted that the two-photon excitation rate is given by $p_{peak}^2 * t_{pulse}$[30,45], so even with longer pulses similar TPM signal levels can be achieved if the duty cycle of the pulses is the same[30,45]. In SLIDE, the repetition rate is typically 88 MHz and thus duty cycles are normally higher than in femtosecond TPM

systems (i.e. lower peak powers in SLIDE). In principle, however, same peak powers and thus same excitation rates can easily be achieved with SLIDE by simply digitally programming the required pulse repetition rate. This powerful pulse on demand feature of SLIDE can be used to trade speed for sensitivity when imaging dim fluorophores, i.e., with low two-photon absorption cross sections.

Interestingly, the excitation rates can also be scaled up by employing shorter pulses. Currently, the electronic pulse generation of 65 ps is the practical limiting factor, as the EOM bandwidth permits much shorter pulses. Furthermore, dispersion-based compression could be harnessed to achieve shorter pulses in the single-digit picosecond range[41,46]. To this end, a large time-bandwidth product is required, which is however achievable with FDML lasers[37].

On the other hand, SLIDE profits from the longer pulses, since to increase photon numbers per single pulse it is advantageous to increase the pulse length rather than the peak power in order to avoid supra-quadratic scaling effects like photobleaching[32,33] and photodamage[35,45]. This advantage of longer pulses was already reported before[30,35,45,47] and was also discussed above. Another factor which reduces photodamage is the longer wavelengths of 1060 nm employed here[48]. Lastly, even though pulse energies become larger with longer pulses, photothermal effects on the sample are negligible in SLIDE microscopy as they are insignificantly small[49] and dissipate in between frame scans.

The fast frame-scans also lead to a low effective repetition rate per pixel of only 1 kHz (i.e. half the bi-directional frame-rate) in SLIDE. These low repetition rates per pixel were reported to lead to advantageous relaxation of even long lived triplet states[32], which can significantly increase photon numbers and thus further helps to speed up FLIM imaging rates.

The SLIDE system presented here has an excitation wavelength around 1060 nm, however future laser developments will target excitation wavelengths at 780 nm for autofluorescence applications[4,5] and 940 nm for GCaMP-based imaging[13].

Regarding sensitivity, Figs. 3 and 4 also show that SLIDE has sensitivity down to the single molecule level, so even samples where only a single fluorescent emitter is excited in the focal volume can be successfully imaged with SLIDE. In this sparse signal case, required photon counts can be reached either by spatial binning or, in applications where high spatial resolution FLIM is required[5], by phase-locked detection and averaging. In SLIDE, frame averaging instead of pixel averaging can be performed, leading to low effective pulse repetition rate per pixel which can help increase signal levels[6] (see above). The analogue detection for FLIM is furthermore compatible with existing two photon microscopes and can significantly increase FLIM imaging speeds[19–21,23].

In conclusion, in this manuscript we presented the concept of high-speed SLIDE microscopy along with the experimental implementation and application in Two-Photon imaging flow cytometry with two imaging modalities, TPM and FLIM. High speeds of 342 kHz line scanning rates and 88 MHz pixel rates were presented. We presented high quality fluorescence lifetime imaging flow cytometry at very high speeds (>10,000 events per second). We believe that this high throughput and multi-modality enabled by SLIDE microscopy can lead to new insights into rapid biological processes and detection of rare events in applications like liquid biopsy or rare circulating tumour cell detection[50]. Further, the fiber-based setup of the SLIDE microscopy system may enable an endoscopic application to overcome the relatively shallow penetration depth of optical microscopy.

## Methods

**Wavelength-to-space mapping.** Upon exiting the single-mode fibre, the light was collimated using an $f = 37$ mm lens (Thorlabs collimator F810APC-1064) followed by a beam-expander ($f = 100$ mm and $f = 150$ mm; Thorlabs LA1509-C and LA1433-C). This results in a beam diameter of 11.5 mm filling the 60× microscope objective aperture. The grating (Thorlabs GR25-1210) was positioned at a 30° angle (close to blaze angle), such that the first order was reflected at almost the incident direction in order to minimize ellipticity of the first-order diffraction beam. At 1200 lines/mm the grating only produced a 0 and +1 diffraction order and the first order power was maximized by adjusting the polarization on a polarization control paddle. The blazed grating ensured >80% power in the first order. The grating resolution is calculated to 67 pm. This fits well to the instantaneous linewidth of the FDML, which was measured for a single pulse to be 56 pm (cf. Fig. 2).

**Microscopy setup.** Two lenses were used to relay image the beams onto a galvanometric mirror for y-axis scanning. The galvo mirror (Cambridge Technology CT6215H) was driven synchronously, producing 170 lines at 2.012 kHz. A high NA, oil immersion microscope objective was used (Nikon Plan Apo 60× NA 1.4 oil) or a 40× water immersion objective (Nikon N40×-NIR - 40× Nikon CFI APO NIR Objective, 0.80 NA, 3.5 mm WD). The field-of-view (FOV) was determined by inserting a resolution target and recording the reflected excitation light on a CCD camera installed in the microscope, which was sensitive to the 1064 nm excitation light (cf. Fig. 3). A dichroic mirror (Thorlabs DMSP950R) in combination with an additional short-pass optical filter (Semrock FF01-750) transmits the Epi-generated signals to a hybrid photodetector (HPD, Hamamatsu R10467U-40) with high quantum efficiency (45%). The high time resolution of the HPD in combination with a fast digitizer (up to 4GS/s) leads to a fast IRF of only 1026 ps, measured by detecting the instantaneous signal of SHG in urea crystals (cf. Fig. 2).

**Digitally synthesized waveforms.** The whole system is driven by an arbitrary waveform generator (AWG, Tektronix AWG7052). This AWG provides all digitally synthesized driving waveforms, driving the FDML laser (Fabry-Pérot Filter waveform and 50% modulation of SOA for buffering), the galvo-mirror and also generates an external sample clock signal for the digitizer. The waveforms are digitally programmed and enable flexibility on the number of pulses per sweep, pulse width, pulse shape, pulse pattern and thus also the image sampling density.

**Digitizer.** As digitizers, either an oscilloscope (Tektronix DPO71604B) at 3.125 GSamples/s or a streaming ADC card (Innovative Integrations Andale X6GSPS and Alazartech ATS9373) with synchronously driven sample clock at 3196 MHz or 3940 MHz, respectively, were employed. The external sample clock was employed such that the data acquisition runs synchronously to the FDML laser and the pulse modulation and ensures sample-accurate fitting. In order to acquire large data sets, a streaming ADC in combination with a RAID-SSD array was employed to store the data.

**Flow cytometry.** In the flow cytometry recording, the flow-rate was set by two fundamental properties, namely the fluorescence lifetime and the imaging diffraction limit. The lifetime limits the repetition rate to ~100 MHz, while the diffraction limit is sampled at ~380 nm. Consequently, we employed 88 MHz repetition rate at 256 pulses per 2.92 µs linescan rate and 100 µm FOV. The flow rate was equally set to sample each line at 380 nm, i.e., 380 nm/2.92 µs = 0.13 m/s. The scale bars in the flow cytometry images were generated using the known 10 µm size of the Red-species bead to calibrate the actual flow speed. The Red bead was sampled with 18 lines, calculating to a line spacing of 556 nm. Using the line scan rate of 342 kHz, this calculates to a flow speed of ~0.2 m/s.

**Data processing.** For precise measurements, a deconvolution with the IRF was conducted in order to extract the fluorescent lifetimes (e.g. in Fig. 3 and Supplementary Fig. 4). However, this is time consuming, so for faster processing and qualitative results a tail-fitting algorithm was used. Often time, different species need to be discerned so a qualitative value is sufficient. The presence of multi-exponential behaviour was tested by checking for kinks in the slope of the logarithmic plot or checking for deviations between the mono-exponential fit and the measurement data (see e.g. Supplementary Fig. 4 where the mono-exponential fit yields a reliable fit). Photon numbers were calculated by dividing the area under the curve by the area of a single-photon event. For all images, the data was processed and images created in LabVIEW. The 2P-FLIM images were generated as HSL-images, where Hue was given by the lifetime-values, lightness by the integrated TPEF signal and saturation always set to maximum, i.e., the same for all images. For the TPEF images, the "Red Hot" or "Royal" look-up tables were applied in ImageJ. The plots were generated in GNUPlot and the figures produced using Inkscape.

**Samples.** The Pollen grain samples were ordered from Carolina (B690 slide). The fluorescent beads were ordered from ThermoFisher Scientific (# F8825, F8839, F8841, F8843, and F21012).

**Experimental model systems: Euglena cell culture and fermentation.** The *E. gracilis* cells used in the study are *Euglena gracilis Z* (NIES-48) strain procured

from the Microbial Culture Collection at the National Institute for Environmental Studies (NIES), Japan. *E. gracilis* were cultured heterotrophically in 500 mL flasks using Koren-Hutner (KH) medium at a pH of 3.5[51]. The cell cultures were maintained at 23 °C with a shaking rate of 120 strokes/min under continuous illumination of 100 μmol photons m$^{-2}$ s$^{-1}$. Cells were subjected to anaerobic fermentation to induce lipid accumulation. The fermentation was performed on cells in stationary phase by bubbling with nitrogen gas and incubating the flasks in the dark for three days.

**Nile red staining of intracellular lipid droplets**. The Nile red stock was prepared by dissolving original dye powder (Sigma) into 4 mL dimethyl sulfoxide (DMSO) to achieve a concentration of 15.9 mg/mL (50 mM). The stain stock solution was diluted 1,000 times with distilled water before use. The *E. gracilis* cells in the culture medium were washed with distilled water and resuspended in distilled water with a final concentration of $2 \times 10^6$ cells/mL. We mixed 15.9 μg/mL of nile red solution with *E. gracilis* cell suspension solution at a volume ratio of 1:1, which was followed by gentle vibration and incubation in the dark for 10 min. The final concentration of Nile red and *E. gracilis* cells were 7.95 μg/mL and $1 \times 10^6$ cells/mL, respectively. The *E. gracilis* cells were washed three times with distilled water and centrifugation ($2000 \times g$, 1 min). The cells were resuspended in distilled water and protected from light prior to imaging.

**Experimental model systems: mouse brain imaging**. All experiments followed the U.S. National Institutes of Health guidelines for animal research, under an animal use protocol (ARC #2007-035) approved by the Chancellor's Animal Research Committee and Office for Animal Research Oversight at the University of California, Los Angeles. Experiments in Supplementary materials used FVB.129P2 wild-type mice (JAX line 004828) injected with AAV1-pCAG-tdTomato (Addgene plasmid # 59462) into the right primary visual cortex. For the viral injections, mice were anaesthetized with isoflurane (5% induction and 1.5% maintenance) and placed on a stereotaxic surgery frame. Next, a small burr hole was drilled over the right primary visual cortex and ~50 nL of adeno associated virus (AAV) was injected 100–200 microns into the cortex. Following injections, the exposed skull was covered with dental cement. The mice were returned to their home cages and two weeks after viral injection, mice expressing TdTomato were perfused intracardially with 4% paraformaldehyde in phosphate buffer and their brains were extracted and glued on a petri dish, which was filled with PBS buffer for imaging on the inverted microscope. Neurons expressing TdTomato were imaged in Layer 2/3 at a depth of ~150–250 μm below the dura using a long working range 40× microscope objective (Nikon N40×-NIR - 40× Nikon CFI APO NIR Objective, 0.80 NA, 3.5 mm WD).

## Data availability
The data that support the findings of this study are available from the corresponding author upon reasonable request.

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

## Acknowledgements

This research was sponsored in part by the National Institutes of Health grants 5R21GM107924-03 to B.J. and C.P.-C. and R21EB019645 to B.J., Cal-BRAIN grant 350050 (California Blueprint for Research to Advance Innovations in Neuroscience) to B.J. and C.P.-C., as well as grant W81XWH-14-1-0433 (USAMRMC, DOD), and NIH NICHD grant R01 HD054453 to C.P.-C. and by ImPACT Program of the Council of Science, Technology and Innovation (Cabinet office, Government of Japan) to C.R. and D.D.C. Sebastian Karpf gratefully acknowledges a postdoctoral research fellowship from the German Research Foundation (DFG, project KA 4354/1-1), the Juniorprofessorship with financial support by the state of Schleswig-Holstein (Excellence chair program by the universities Kiel and Luebeck) and funding from the Deutsche Forschungsgemeinschaft (DFG, German Research Foundation) under Germany's Excellence Strategy – EXC 2167-390884018.

## Author contributions

S.K. conceived the idea, built the system and conducted the experiments. C.R. and D.D.C. provided the Euglena gracilis cells. C.R. and S.K. conducted the Euglena flow measurements. A.G, W.Z., A.S. and C.P.C. provided the mouse brain sample. C.P.C. and S.K. conducted the mouse brain imaging measurement. S.K. and B.J. conceived the digital image processing capabilities. B.J. and S.K. performed system analysis and wrote the manuscript. B.J. supervised the research.

## Competing interests

The authors declare no competing interests.
