## [Peer Review File · Nature Communications]

Reviewers' comments:

Reviewer #1 (Remarks to the Author):

In this manuscript (NCOMMS-19-32315), Karpf et al describe a spectro-temporal lifetime Imaging by diffracted excitation (SLIDE) microscopy approach for high-speed two-photon fluorescence intensity, fluorescence lifetime imaging microscopy (FLIM), and second harmonic generation (SHG) imaging at kilohertz frame rates. In this novel imaging approach, the authors use pulse-modulated, rapidly wavelength-swept lasers and inertia-free beam steering through angular dispersion. Using SLIDE with a high bandwidth, single-photon sensitive detector, the authors have recorded the fluorescent lifetimes of fluorescence beads and other bright fluorophores (with high fluorescence quantum yield) at 88 million pixels per second. As a proof of principle, the authors also combined their diffraction-limited, Two-Photon fluorescence and FLIM imaging with flow cytometry for image-assisted cell sorting.

The proposed experimental SLIDE approach overcomes the inherent challenges associated with conventional two-photon, laser-scanning fluorescence and FLIM such as the limited data acquisition rate (number of frames per second) at a descent signal-to-noise ratio and good spatial resolution for reliable FLIM data analysis. On the other hand, the SLIDE technique has its own limitations such as the required high fluorescence quantum yield.

The manuscript is well written and the results support the stated conclusions by the authors. This is certainly a step forward to push the speed limits of 2P-FLIM towards real-time monitoring on biological systems that are dynamics such as neural activities. However, there are some comments to the authors (below) that need to be addressed by the authors first prior to the recommendation to publish this manuscript.

Comments to the Authors:

(1) In a biological sample that may contains multiple fluorescence markers, each of which may have an absorption band up to 100 nm bandwidth, it is not clear to this reviewer the relationship between the dispersed laser pulses and such broad absorption band of typical molecules in solution or in cells. How this spectral dispersion of the laser pulses may complicate the fluorescence lifetime of each fluorophore per pixel per wavelength? Please elaborate.

(2) With FWHM system response function of 1026 ps and a typical signal-to-noise ration in this SLIDE approach, what is the projected temporal resolution in FLIM measurements under typical conditions? Please elaborate.

(3) Page 7, Line 124: If the authors used tail-fitting approach, does this mean there is no deconvolution required for the measured fluorescence decay per pixel with the system response function? If so, perhaps the authors can provide a range of fluorescence lifetime (i.e., fluorophores) that can be ideally measured with this new approach (i.e., how long lifetime compared with 1026 ps SRF?). Without deconvolution, how is it possible to measure 1.2 ns lifetime if the SRF is 1.026 ns (FWHM) as shown in Figure 4C? Please elaborate.

(4) For significant comparison with the traditional two-photon FLIM, it will be helpful if the authors can provide (1) the fluorescence quantum yield of each fluorophore or nanoparticle used in this proof-of-principle manuscript and (2) what is the literature value of the corresponding fluorescence lifetime of each as compared with SLIDER.

(5) Page 8, Line 163: If the fluorescence quantum yield of green fluorescence protein mutations ranges from 0.5 - 0.8, what is the fluorescence quantum yield range that can be excluded from SLIDE approach? Please elaborate.

(6) Page 9, Line 173: The authors write, "Therefore, biochemically engineered fast and bright fluorescent proteins are needed, which can also be expressed at high abundance without interfering with cellular behaviour, in order to enable kHz frame-rate imaging of neuronal activity for SLIDE or any other fast technique in the future." What is meant by "fast" in this sentence; maturation? What is the required expression level needed for SLIDE without interference with the biological cell machinery?

(7) Parts of comment (6) concerning the expression level seems contradictory with the stated single-molecule sensitivity of SLIDE (Line 187). Please elaborate.

(8) Page 9, Line 177: How did the authors exclude the photo damage caused by the high average laser power used in SLIDE (10-100 mW)?

(9) The authors may want to describe whether SLIDE is compatible with existing two photon FLIM or it requires a new stand alone setup.

Reviewer #2 (Remarks to the Author):

This paper describes a new method for point-scanning multiphoton microscopy, using a rapidly-swept pulsed laser (specifically a Fourier Domain Mode-Locked laser) which is gated using an EOM and amplified to yield a train of ~65ps pulses which are dispersed by a grating. To the best of my

knowledge, this is a novel method for laser scanning - conventionally a Titanium-Sapphire or Yb-fiber oscillator (operating with a pulse repetition rate of ~ 80 MHz) is steered using galvo mirrors, polygonal scanners, acousto-optic deflectors, a spatial light modulator or some other beamsteering / patterning device. Using the laser itself to scan the beam is both elegant and potentially valuable, as it is very fast yet robust. I should state from the start that while I am supportive of this work appearing in Nature Communications (as I believe that high-profile journals should be publishing promising and novel-but-undeveloped techniques, and not just the usual headline-grabbing derivative and intellectually-unimaginative work that only excites researchers unfamiliar with the field) there are a number of significant flaws in the technique that should either be corrected or highlighted for readers to assess themselves. I will discuss these major concerns here:

My first concern is that the bandwidth of the laser is shared amongst all pulses, making the pulse duration abnormally long for multiphoton excitation (due to the time-bandwidth product). Ordinarily a pulse duration of ~ 100 fs at the sample is deemed optimal for multiphoton microscopy, but the pulse in this case is nearly 1000x longer, at ~ 65 ps. Since fluorescence scales inversely with pulse duration, this inefficiency is considerable. While I am impressed that the authors were able to achieve 2P imaging using this system, it doesn't surprise me that high powers and bright samples were necessary, along with significant averaging of frames / pixels needed to resolve the fluorescence lifetime. At the very least the authors should compare image brightness with their system versus a conventional femtosecond scanning system, if not redesign the laser to operate with wider bandwidth and hence shorter pulses.

A second concern is related to the first - multiphoton FLIM is perhaps one of the techniques which least benefits from high scanning speeds. FLIM requires several hundred photons to accurately fit a single exponential decay, and several thousand if the decay is multi-exponential (as it is in many biological samples). Since it is not uncommon to gather only a few tens of photons or fewer per excitation pulse, it is necessary to average many frames or pixels together to resolve the lifetime. Indeed, the authors observed precisely this - they required 200 frame averages, along with 3x3 or even 9x9 spatial binning to capture enough photons for FLIM. As such I really cannot support the claim that this represents 88 MHz lifetime pixel rates (line 40, 197 and elsewhere) - it is simply not possible using this system to capture FLIM data at 88 million pixels per second even for very bright samples, let alone more conventional samples. As a further aside, claiming the world's fastest FLIM system has further difficulties, as it would be necessary to compare with widefield gated FLIM imaging as well. While these might not be able to resolve lifetime on the order of nanoseconds, they can however do so for millions of pixels in parallel, which can be a major advantage over single-point scanning systems.

Following on from the previous concern, the authors have also failed to cite significant achievements in the literature from researchers working on high-speed multiphoton and lifetime imaging. [Giacomelli et al., Biomed Opt Express 6(11):4317, 2015], [Bowman et al., Nature Comms 10:4561, 2019], [Poland et al., Biomed Opt Express 6(2):277, 2015], [Krstajic et al., Opt Lett 40(18):4305 2015] and [Raspe et al., Nature Methods 13:501, 2016] would all seem to be relevant to this paper; I'm sure the authors can find more. A good literature review is at the heart of good research, and it is

important for the wide readership of Nature Communications that this work is placed in its proper context.

The drive for high pixel throughput is laudable, but one of the approaches commonly used to achieve this is to increase the laser power to produce more fluorescence photons per pulse. Since this degrades the resolution of the system, it would be important to quantify the spatial resolution in X,Y and Z, by imaging some sub-diffraction-limited fluorescent beads for example.

There are also more minor concerns which should be addressed but are not as critical. These are as follows:

Line 27 is debatable - the I^2 dependence of multiphoton excitation only favours single-point scanning if the focus remains unsaturated; because saturation is comparatively easy to achieve with the ~ 1 -5W available from modern femtosecond lasers, optimal use of laser power actually favours scanning of multiple spots simultaneously (a technique known as Multifocal Multiphoton Microscopy) and in the extreme case, favours excitation of a whole field of view at once (also known as Temporal Focusing). If the user is trying to image deep into tissue then these parallel techniques are unfavourable, owing to crosstalk between the foci, but this also doesn't particularly support the authors' need to justify their rapid imaging, as the number of fluorescence photons captured when scanning deep into tissue is much lower than at the surface, and hence pixel dwell times are not limited by the speed of the scanning system.

The methods section should contain a concise description of the steps required to replicate the authors' results. Currently there is a great deal of material that would be better placed in the main body of the text, or at the very minimum in Supplementary Information. Furthermore, all parts in the system should be described - the manufacturer and the model number are required for each part. This is particularly important for the galvo scanner, which is an integral part of the high-speed scanning system at the heart of this paper.

There has been no attempt to determine whether the FLIM results are accurate - at the bare minimum, comparisons using a sample with a known fluorescence lifetime should be performed. There is a sentence (line 73) to say that this has been done, but no evidence is provided. In particular, photomultiplier tubes can have saturation effects at high gain (recording one photon decreases the probability of recording one within the next few hundred picoseconds) which might be manifested as a bias in the lifetime measurement when directly recording the fluorescence decay.

The fact that the laser cannot be tuned easily over a wide range of wavelengths is a limitation for practical use, and this should be mentioned in the text. ~ 1060 nm excitation only permits excitation of dyes that lie towards the red end of the spectrum, leaving a great many green and blue dyes

unusable. Since many important proteins (GCaMP, ChR2, GFP etc.) are green, this is a strong limitation for practical use of the technique and should be mentioned.

Line 118 - the fact that the beads are approximately the same size as cells does not make them a good proxy for cells. Fluorescent beads are much brighter than even the brightest cells, and it is unlikely that imaging a 15um bead will provide useful information on the applicability of the technique to biological imaging.

It would be good practice to release all code, data and supporting information in an open source repository. This would aid other researchers in reproducing this work.

Line 441 - 'Mouse' is misspelled. Furthermore, it would be good to include or reference a detailed protocol for the transfection and other animal procedures.

The authors should be prepared to justify their choice of single-exponential fitting for all their lifetime measurements. Many fluorescent proteins are better fitted with a multiexponential distribution, so it is important to be sure that a monoexponential model is truly the best choice.

I conclude by reiterating my view that this is the kind of research which should be published in high-profile journals such as Nature Communications, but a significant amount of extra work is needed before it is ready, as well as a more realistic and context-aware assessment of the capabilities of the new system. I look forward to reviewing a more polished version of this manuscript.

Response to Reviewers

All citations/reference numbers stated refer to the original manuscript submission, not to the revised manuscript. Thus, numbers may vary to the revised manuscript as new references were added to the revised manuscript.

Reviewer #1 (Remarks to the Author):

In this manuscript (NCOMMS-19-32315), Karpf et al describe a spectro-temporal lifetime Imaging by diffracted excitation (SLIDE) microscopy approach for high-speed two-photon fluorescence intensity, fluorescence lifetime imaging microscopy (FLIM), and second harmonic generation (SHG) imaging at kilohertz frame rates. In this novel imaging approach, the authors use pulse-modulated, rapidly wavelength-swept lasers and inertia-free beam steering through angular dispersion. Using SLIDE with a high bandwidth, single-photon sensitive detector, the authors have recorded the fluorescent lifetimes of fluorescence beads and other bright fluorophores (with high fluorescence quantum yield) at 88 million pixels per second. As a proof of principle, the authors also combined their diffraction-limited, Two-Photon fluorescence and FLIM imaging with flow cytometry for image-assisted cell sorting.

The proposed experimental SLIDE approach overcomes the inherent challenges associated with conventional two-photon, laser-scanning fluorescence and FLIM such as the limited data acquisition rate (number of frames per second) at a descent signal-to-noise ratio and good spatial resolution for reliable FLIM data analysis. On the other hand, the SLIDE technique has its own limitations such as the required high fluorescence quantum yield.

The manuscript is well written and the results support the stated conclusions by the authors. This is certainly a step forward to push the speed limits of 2P-FLIM towards real-time monitoring on biological systems that are dynamics such as neural activities. However, there are some comments to the authors (below) that need to be addressed by the authors first prior to the recommendation to publish this manuscript.

Comments to the Authors:

(1) In a biological sample that may contains multiple fluorescence markers, each of which may have an absorption band up to 100 nm bandwidth, it is not clear to this reviewer the relationship between the dispersed laser pulses and such broad absorption band of typical molecules in solution or in cells. How this spectral dispersion of the laser pulses may complicate the fluorescence lifetime of each fluorophore per pixel per wavelength? Please elaborate.

Author's response:

We thank the reviewer for the detailed analysis of our manuscript. For the spectral dispersion used in the SLIDE microscope, it is important to note that it serves solely to achieve a fast line scan. Other than that, the situation is similar to regular line-scanning microscopy. Each pixel is illuminated with a quasi-monochromatic light (56pm spectral width at ~1060nm). As you stated above, most absorption bandwidths are broad, so the wavelength change over a line can be neglected (only 12nm relative to 1060nm or ~1%) against typical absorption bandwidths (~100nm). Hence, apart from the inertia-free line scanning achieved through the wavelength-sweep, the excitation light for each pixel can be regarded as the same colour. Importantly, in almost all cases the fluorescence emission is independent on the excitation wavelength (Kasha's rule), as transitions corresponding to the Stokes shift are much faster than the fluorescence lifetime. Thus, the spectral dispersion of the laser pulses does not influence the fluorescence nor the lifetime of each pixel.

We added to the manuscript the following information. Line 61 ff. now reads:

"It shall be noted the used spectral bandwidth of 12nm (Fig. 2E) lies well within most Two-Photon absorption bandwidths, so after absorption and excited state relaxation (Stokes shift), the fluorescence emission is equivalent for all pixels along the whole line (Kasha's rule)."

(2) With FWHM system response function of 1026 ps and a typical signal-to-noise ration in this SLIDE approach, what is the projected temporal resolution in FLIM measurements under typical conditions? Please elaborate.

Author's response:

This is a very good question. The precise value of the fluorescence lifetime can in theory be extracted from the measured temporal curve through deconvolution with the instrument response function. By that, even lifetimes much shorter than the instrument response function can be measured, similar to super-resolution methods in microscopy where precise localization much beyond the PSF (or diffraction-limited spot size) can be determined. In practice, however, this precise extraction of the lifetime is hindered by limited fluorescence activity or integration time (leading to limited SNR), timing jitter etc., so the determination of the lifetime will be instrument-, sample- and measurement dependent. In this proof-of-principle manuscript it is safe to say that lifetimes down to the IRF can be reliably determined and relative changes in lifetime, e.g. from 2ns to 3ns can also readily be distinguished. In addition, lifetimes in this range determined with this fast analogue approach compare well to literature values (see Karpf et al, BOE 2016, ref. 8). Thus, the lifetime here can be used to gain an additional contrast by discerning qualitative lifetime differences. In the future, in order to extract quantitative values down to very short lifetime values, a more in depth analysis together with fitting robustness assessment and comparison measurements with state-of-the-art FLIM systems is necessary. This analysis will be conducted in subsequent works, which is beyond the scope of this initial report presenting the new SLIDE microscopy technique together with high-speed Two-photon and 2P-FLIM imaging flow cytometry with sub-cellular resolution and qualitative lifetime contrast.

In summary, although in principle lifetimes below the instrument response function should be measurable with this approach, here only qualitative analysis is reported. A detailed analysis will follow with future work.

(3) Page 7, Line 124: *If the authors used tail-fitting approach, does this mean there is no deconvolution required for the measured fluorescence decay per pixel with the system response function? If so, perhaps the authors can provide a range of fluorescence lifetime (i.e., fluorophores) that can be ideally measured with this new approach (i.e., how long lifetime compared with 1026 ps SRF?). Without deconvolution, how is it possible to measure 1.2 ns lifetime if the SRF is 1.026 ns (FWHM) as shown in Figure 4C? Please elaborate.*

Author's response:

The tail-fitting approach yields reasonable results for lifetimes that are much longer than the half width half maximum (HWHM), or 512ps in this case (see https://www.tcspc.com/doku.php/howto:lifetime_fitting_using_the_flim_analysis).

Arguably, the 1.2ns measurement from Figure 4C is too short to be reasonably extracted from the data just by tail-fitting. In fact, the fit does not show a very good agreement with the experimental data. Here, however, only a qualitative comparison between the five species of fluorescence beads was the aim, as this application provides a new possibility of using FLIM imaging in flow cytometry as additional physical information channel. In order to push this approach to a real-time compatible application, both the flow speed as well as the lifetime extraction algorithm shall be speed efficient. This will also limit the photon count numbers so quantitative analysis appears unrealistic. Hence, here a deconvolution and a detailed lifetime extraction would have been prohibitively slow computationally.

In contrast, for the mouse brain imaging shown in supplementary figure 4 a deconvoluted lifetime extraction with a fit in high accordance to the experimental data is presented. The extracted lifetime gained in this way (esp. including averaging) will have a much more quantitative and interpretable value than the qualitative nature of the flow measurements presented in Figs. 4 and 5.

In practice, however, many applications in flow cytometry will not require the absolute lifetime value but rather a qualitative, relative value in order to successfully distinguish different species or track different lifetimes. Thus, we envision a high-speed FLIM-imaging based flow cytometry application, e.g. in cell sorting enabled by SLIDE flow cytometry.

We added to the manuscript the following information. Line 71 ff. now reads:

"We confirmed the recorded lifetime values for Rhodamine 6G to literature values in previous works to confirm the validity of the analogue lifetime detection approach."

Further, line 123 ff. now reads:

"For this application, tail-fitting was applied assuming lifetimes significantly longer than half of the IRF. This approach will achieve fast fitting and still yield reliable qualitative lifetime contrast in order to distinguish different lifetimes."

(4) *For significant comparison with the traditional two-photon FLIM, it will be helpful if the authors can provide (1) the fluorescence quantum yield of each fluorophore or nanoparticle used in this proof-of-principle manuscript and (2) what is the literature value of the corresponding fluorescence lifetime of each as compared with SLIDER.*

Author's response:

Although the precise determination of quantum yield and lifetime is difficult as they are also solvent-dependent, here is the result of literature search. It shall serve as an indicator for approximate values: Figs. 3 & 5, the autofluorescent chlorophyll has a quantum yield of 0.1-0.25 (Forster, Livingston, J.Chem.Phys. 20, 1952) and fluorescence lifetime of 0.5ns to 0.7ns

(Singhal, Rabinowitch, Biophys. J. 9, 1969) while Nile Red has a reported quantum yield of 0.7 (Sackett, Wolff, Anal. Biochem. 167, 1987) and a lifetime of ~3-4ns (Cser et al., Chem.Phys.Lett. 360, 2002). These values are in good agreement with our measured lifetimes.

Regarding the beads in Fig. 4, bead I was stained with crimson which has a reported quantum yield of 0.23 (Strack et al., Biochemistry 48, 2009) and lifetime of ~1ns (Lin et al., Jap.J.Appl.Phys. 52, 2013). Bead II, the smallest bead, was stained with Nile Red, whose values are already written above. The lifetime of ~3-4ns is shown in yellow and agrees well. Bead III "orange" has a reported lifetime of 4.6ns (So et al., SPIE Vol. 2137, 1994) and unknown quantum yield, as Thermo-Fisher does not disclose which fluorophore they use. The same holds for bead IV "red" and V "red-orange", so unfortunately we could not find lifetime nor quantum yield values for these proprietary beads.

The reported values for tdTomato (Supp. Figs.1-4) are quantum yield of 0.69 (<http://www.fluorophores.tugraz.at/substance/269>) and lifetime of 3.4ns (Kumar et al., OL 34, 2009). Our measured lifetime is significantly shorter than this literature value, which could be explained by an influence of the applied PBS buffer (Pliss et al., ACS Chem. Biol. 7, 2012). However, a detailed analysis of this effect is outstanding.

The Pollen grain (Supp.Fig. 5) was stained either with fast green or with phloxine B (e-mail from Carolina Biological supplies). There is a literature report of FLIM imaging of a similar Pollen grain reporting 0,8ns (Grant et al., OL 30, 2005) which agrees well with our fitted lifetime of 1ns using tail-fitting. However, we could not find a quantum yield reference. We prefer to not include a quantitative lifetime analysis in the paper, as a more rigorous and complete assessment has to be conducted first (including comparison measurements with TCSPC systems, which we had no access to). In this proof-of-principle manuscript, only qualitative lifetime assessment was conducted showing however promising capability for a future quantitative analysis.

(5) Page 8, Line 163: If the fluorescence quantum yield of green fluorescence protein mutations ranges from 0.5 - 0.8, what is the fluorescence quantum yield range that can be excluded from SLIDE approach? Please elaborate.

Author's response:

As stated above, we measured fluorophores with quantum yield of 0.1 and higher. However, we believe that this is not the limiting factor, as we still could have increased the illumination laser power without observing any damage to the samples. However, in the results obtained from the genetically expressed tdTomato fluorescent proteins we observed a saturation of the fluorophores in the sample. Thus, we wanted to highlight that a higher number of fluorescent photons per pulse would greatly benefit SLIDE in obtaining its maximum speed. In traditional TCSPC FLIM systems, only less than one photon per excitation pulse is required/wanted. In the analogue approach of SLIDE much greater photon numbers can be reached thus greatly boosting FLIM speeds (see Figs. 3-5). Of course, the quantum yield should be kept as high as possible in order to permit fastest possible measurement times, however also two-photon absorption cross section, nonlinear damage mechanisms, photobleaching etc. will determine the maximum achievable speed. Also, there are much different requirements for TPM and 2P-FLIM. In the case of supplementary figure 4, high-speed TPM recordings were possible at 1kHz frame-rate yet FLIM imaging was achievable at 10Hz frame-rate.

(6) Page 9, Line 173: The authors write, "Therefore, biochemically engineered fast and bright fluorescent proteins are needed, which can also be expressed at high abundance without interfering with cellular behaviour, in order to enable kHz frame-rate imaging of neuronal activity for SLIDE or any other fast technique in the future." What is meant by "fast" in this sentence; maturation? What is the required expression level needed for SLIDE without interference with the biological cell machinery?

Author's response:

By fast, we mean imaging at the biological millisecond timescale of action potentials, i.e. >1000 frames per second. This is meant in this paragraph with "kHz frame-rate". For a minimum image size of 200x200 pixels and a laser scanning system this calculates to 40 millions pixels per second. In order to achieve this fast imaging speed, a high number of photons should be available within such a short pixel dwell time. This is a requirement for any fast system, not just for SLIDE. Our experience with GCaMP fluorescence lead us to the conclusion that the red-shifted tdTomato or jrGeCO didn't work as well. For now, the SLIDE system operates at 1060nm excitation wavelength so we had to use these available fluorescent proteins. However, we want to shift the SLIDE technique also to other wavelengths in order to be able to image GCaMP-based neuronal activity leading to higher SNR levels.

Further, current calcium-sensitive dyes don't show the required speeds, but would scale for a volumetric imaging approach with SLIDE, e.g. volumetric imaging with 200 z-stacks at 10Hz. However, we hope that biochemically engineered voltage-sensitive dyes with fast speeds (see e.g. Gong et al., Science 350, 2015, ref. 2) will be available soon at high brightness in order to make fast imaging at kHz frame-rate possible.

Regarding expression levels, there is no quantitative number to be stated as of now but a higher fluorophore concentration would have greatly improved imaging results in sup.figs. 2-5, especially for FLIM speeds.

(7) Parts of comment (6) concerning the expression level seems contradictory with the stated single-molecule sensitivity of SLIDE (Line 187). Please elaborate.

Author's response:

As stated above, TPM imaging was possible as even a single photon achieves an SNR >1. However, when aiming at neuronal activity imaging either through calcium concentration induced fluorescence intensity change, voltage-sensitive intensity change or FLIM lifetime changes, many photons are required per pixel in order to detect a ΔF . This adds to the required integration time and is a limit on the biological side, not on the instrumentation side. SLIDE has sufficient sensitivity and dynamic range to achieve this feat, yet here this speed was hindered by the available numbers of fluorescent proteins.

(8) Page 9, Line 177: How did the authors exclude the photo damage caused by the high average laser power used in SLIDE (10-100 mW)?

Author's response:

There is a great number of empirical evidence that the current limiting factor in MPM stems from nonlinear processes (see e.g. Hopt&Neher, Biophysical Journal 80, 2001, ref. 15), so actually the high peak power of ultrashort pulses is the limiting factor, not the average power. As our pulses are longer than typical 100fs pulses, we worked with much lower peak powers, thus drastically lower probability of nonlinear damages occurring. Linear damage through heating can be excluded due to the fast scanning (see Schönle&Hell, OL 23, 2005, ref. 16).

This observation is accordance with other reports (see e.g. Hänninen&Soini&Hell, J.Microscopy 176, 1994).

(9) The authors may want to describe whether SLIDE is compatible with existing two photon FLIM or it requires a new stand alone setup.

Author's response:

SLIDE comprises two main aspects to achieve fast FLIM imaging: 1) the SLIDE microscope with the FDML-MOPA laser, the inertia-free beam scanning and spectro-temporal encoding and 2) the fast analogue detection. For faster FLIM, already the analogue detection can yield significant speed improvements over TCSPC approaches, as was already reported by e.g. Karpf et al. US9851303B2, Dow et al., OL 40, 2015, Giacomelli et al., BOE 4317, 2015 or Eibl et al., BOE 8, 2017. This analogue detection in combination with fast digitizer cards will be readily integrated with an existing two-photon system to achieve fast FLIM imaging up to video rate, i.e. limited by the scanner speed as in conventional two photon systems.

We included a passage (including references) to the manuscript, line 196: *“The analogue detection for FLIM is furthermore compatible with existing two photon microscopes and can significantly increase FLIM imaging speeds.”*

Reviewer 2:

“This paper describes a new method for point-scanning multiphoton microscopy, using a rapidly-swept pulsed laser (specifically a Fourier Domain Mode-Locked laser) which is gated using an EOM and amplified to yield a train of ~65ps pulses which are dispersed by a grating. To the best of my knowledge, this is a novel method for laser scanning - conventionally a Titanium- Sapphire or Yb-fiber oscillator (operating with a pulse repetition rate of ~80MHz) is steered using galvo mirrors, polygonal scanners, acousto-optic deflectors, a spatial light modulator or some other beamsteering / patterning device. Using the laser itself to scan the beam is both elegant and potentially valuable, as it is very fast yet robust. I should state from the start that while I am supportive of this work appearing in Nature Communications (as I believe that high-profile journals should be publishing promising and novel-but-undeveloped techniques, and not just the usual headline-grabbing derivative and intellectually-unimaginative work that only excites researchers unfamiliar with the field) there are a

number of significant flaws in the technique that should either be corrected or highlighted for readers to assess themselves. I will discuss these major concerns here.”

Author’s response:

Reviewer 2 is of the opinion, that some flaws exist in this manuscript and should “either be corrected or highlighted for readers to assess themselves”. We thank the reviewer for finding and highlighting these points and thereby improving the quality of the manuscript. We took utmost care in addressing these issues in the manuscript:

- 1) *“My first concern is that the bandwidth of the laser is shared amongst all pulses, making the pulse duration abnormally long for multiphoton excitation (due to the time-bandwidth product). Ordinarily a pulse duration of ~100fs at the sample is deemed optimal for multiphoton microscopy, but the pulse in this case is nearly 1000x longer, at ~65ps. Since fluorescence scales inversely with pulse duration, this inefficiency is considerable. While I am impressed that the authors were able to achieve 2P imaging using this system, it doesn’t surprise me that high powers and bright samples were necessary, along with significant averaging of frames / pixels needed to resolve the fluorescence lifetime. At the very least the authors should compare image brightness with their system versus a conventional femtosecond scanning system, if not redesign the laser to operate with wider bandwidth and hence shorter pulses.”*

Author’s response:

We thank the reviewer for this discussion of the longer pulses. Indeed, a 100fs laser at 80 MHz repetition rate seems the ideal choice for two-photon imaging, since it provides optimal repetition rate for most fluorescence lifetimes and very low duty cycle, thus high peak power to make use of the quadratic peak power scaling of two photon absorption. However, the argument that “fluorescence scales inversely with pulse duration” is not entirely correct, as there is also the repetition rate to the second power in the denominator and the average power squared in the numerator (see the original Denk paper from 1990). The former statement assumes a fixed average power and, more importantly, a fixed repetition rate. This is the case with most femtosecond systems, as rep rates are typ. 80 MHz and average powers for imaging are around 1-50mW, just below damage thresholds. However, by looking at the formula, one finds that if one would have the option to change the rep rate, only a 30-fold reduction in rep rate would lead to the same number of absorbed photon pairs per pulse! Taking further into account the photon numbers per pixel dwell time, the repetition rate should be further reduced to lead to the same factor 1000. In other words, the formula from the Denk paper can be rewritten to show that the two-photon signal scales with the peak power squared and linear in pulse length ($p_{\text{peak}}^2 * t_{\text{pulse}}$). We used this in our 2016 paper [ref. 8] where we showed that, at same peak power, a longer pulse will generate linearly more photons per pulse. This has a big advantage for fast FLIM imaging! See also our paper on single-pulse FLIM, Eibl. et al., BOE 2017.

Is it to be noted, however, that this linear pulse length scaling can be limited by fluorescence saturation (see Supp. Fig. 1 and corresponding discussion).

To fully alleviate the reviewers concern about the pulse length, in our paper from 2016 (ref. 8) we showed a comparison of such a longer pulse system and a state-of-the-art femtosecond system (LaVision Biotech TriMScope). We found that both approaches yielded similar imaging results. Regarding the use of picosecond pulses for TPM we also want to highlight other

pioneering work from the Hell group “Ca²⁺ Fluorescence Imaging with Pico- and Femtosecond Two-Photon Excitation: Signal and Photodamage”, *Biophysical Journal* 77 (1999) (citation added in manuscript).

Therefore, the pulse length of 65ps is currently no significant limitation of the SLIDE system. However, we agree with the reviewer that shorter pulses, perhaps in the single-digit picosecond range, can be advantageous for biomedical imaging. Such short pulses were not obtainable with the current electronics for pulse generation, but future developments will focus on achieving such short pulses, e.g. by faster electronics (the EOM can have up to 40GHz bandwidth) or using sophisticated dispersion compression techniques (see e.g. new reference Obrzud, Lecomte, Herr, *Nat Photonics* 2017). We added a statement and two references to the manuscript to highlight this future direction.

Additionally, we want to address the very valuable point raised by the reviewer of the time-bandwidth limit and the required spectral bandwidth. When we decrease the pulse duration of the pulses to single-digit picoseconds, we also increase the spectral bandwidth of each pulse. This places further requirements on the spectro-temporal bandwidth of the FDML-MOPA laser source. However, FDML-lasers at 1060nm with 143nm spectral bandwidth were already reported [Kolb et al, *BOE* 9 (2018), citation added in manuscript] and the Yb-gain spectral width also permits broader bandwidth and can even be extended by stimulated Raman gain in fibers. In all, these future directions seem promising. To this end, we also have a helpful discussion in our recent manuscript on time-stretch LiDAR with swept FDML lasers (Jiang, Karpf, Jalali, *Nature Photonics* 14, 2020, citation added).

We included the following passage (including references) to the manuscript:

“Regarding signal levels achieved with the SLIDE technique it shall be highlighted that pulse lengths of 65ps are much longer than what is typically employed (~100fs) in TPM. However, longer pulses can still lead to similar signal levels (see also the methods section). In the future, dispersion-based compression could be harnessed to achieve shorter pulses in the single-digit picosecond range. To this end, a large time-bandwidth product is required, which is achievable by FDML lasers.”

2) *“A second concern is related to the first - multiphoton FLIM is perhaps one of the techniques which least benefits from high scanning speeds. FLIM requires several hundred photons to accurately fit a single exponential decay, and several thousand if the decay is multi-exponential (as it is in many biological samples). Since it is not uncommon to gather only a few tens of photons or fewer per excitation pulse, it is necessary to average many frames or pixels together to resolve the lifetime. Indeed, the authors observed precisely this - they required 200 frame averages, along with 3x3 or even 9x9 spatial binning to capture enough photons for FLIM. As such I really cannot support the claim that this represents 88MHz lifetime pixel rates (line 40, 197 and elsewhere) - it is simply not possible using this system to capture FLIM data at 88 million pixels per second even for very bright samples, let alone more conventional samples. As a further aside, claiming the world's fastest FLIM system has further difficulties, as it would be necessary to compare with widefield gated FLIM imaging as well. While these might not be able to resolve lifetime on the order of nanoseconds, they can however do so for millions of pixels in parallel, which can be a major advantage over single-point scanning systems.*”

Author’s response:

We thank the reviewer for the thorough analysis of the implications of our SLIDE systems. Regarding the critique, that “*multiphoton FLIM is perhaps one of the techniques which least benefits from high scanning speeds*”, it was empirically shown that fast scanning leads to a significant increase in fluorescence signal amplitude, as reported by work from Stefan Hell’s group “Major signal increase in fluorescence microscopy through dark-state relaxation”, *Nature Methods*, 2007 (ref. 13) and Arthur Konnerth’s group “Functional mapping of single spines in cortical neurons in vivo”, *Nature*, 2011 (ref. 17). This is probably due to avoiding triplet absorption as the fast scanning and hence low effective pulse repetition rate per pixel leads to a “dark-state relaxation” of the fluorescing molecules before they are excited again. In SLIDE, this effective pixel repetition rate is 1 millisecond, so a sufficiently long time in between pulses for the molecules to return to their ground state. This low effective repetition rate per pixel also further alleviates any thermal load to the sample. In all, this major signal increase, observed at factors up to 25-fold, helps significantly with FLIM imaging.

We changed the discussion in the manuscript to further highlight this advantage for fast-scanning FLIM (line 180):

In fact, the effective repetition rate per pixel of only 1 kHz in SLIDE permits relaxation of even long lived triplet states, which can significantly increase photon numbers and thus further helps to speed up FLIM imaging rates.

As a side note and to further answer the reviewer: This is the first report on the new SLIDE microscope which employs a new scanning mechanism for high-speed inertia-free scanning. We further employed this laser and scanning mechanism on time-stretch LIDAR (Jiang, Karpf, Jalali, *Nature Photonics* 2020) and have also frequency-doubled the wavelength sweep of the FDML-MOPA laser to visible wavelengths (Karpf, Jalali, *OL* 2019). Thus, the SLIDE microscope can be further used for one-photon microscopy which will show better photon number scaling than the inherently confocal Two-Photon excitation microscopy approach. In high-speed point scanning one photon SLIDE microscopy much higher photon numbers can be achieved, leading to even faster FLIM rates. However, this could be practically limited by the maximum photocurrent of the hybrid photodetector. We are investigating this and will report on this in future manuscripts.

>> As such I really cannot support the claim that this represents 88MHz lifetime pixel rates (line 40, 197 and elsewhere) - it is simply not possible using this system to capture FLIM data at 88 million pixels per second even for very bright samples, let alone more conventional samples.

Author’s response:

In the flow measurements and the Euglena still image, we show precisely this: FLIM imaging at 88MHz lifetime pixel rates. This data is shown in Figs. 3, 4 and 5. Therefore, we don’t understand the reviewers critique. We claim what is supported by our experimental data. The only data which employed frame averaging are in the supplementary material. All the figures in the main article support our claim of 88MHz FLIM pixel rate. Further, we tried to make it very clear what photon numbers were achieved alongside which pulse parameters, powers, integration times, averaging etc. was employed. We transparently discussed all the necessary parameters to assess the FLIM imaging performance. Pixel binning is standard procedure in FLIM imaging and is usually not considered when stating a frame-rate or pixel-rate. As a further backup of the claim, in the brain imaging presented in Sup. Figs. 2-5, we show the photon numbers we obtained by 1kHz imaging and discuss that fluorescence saturation lead to the limited photon counts which required further frame averaging in order to generate FLIM images (>100 photons per pixel). High-speed TPM imaging at 1 kHz frame-rate was shown. Only FLIM required frame averaging (due to saturation), so we showed

FLIM imaging of ex vivo neurons at 10Hz, which is still a very fast frame rate for neuronal FLIM and we only present this data in the supplementary material.

>> As a further aside, claiming the world's fastest FLIM system has further difficulties, as it would be necessary to compare with widefield gated FLIM imaging as well. While these might not be able to resolve lifetime on the order of nanoseconds, they can however do so for millions of pixels in parallel, which can be a major advantage over single-point scanning systems.

Author's response:

We added a very recent review article to underline our claim (Liu et al., "Fast fluorescence lifetime imaging techniques: A review on challenge and development"). This review also includes wide-field FLIM technologies as well as other state-of-the-art technologies. Therefore, still to the best of our knowledge, SLIDE is the fastest FLIM imaging system in comparison.

With this statement we just want to highlight that by introducing the SLIDE technique, it is now possible to record FLIM in high throughput imaging flow cytometry, at speeds which would not be possible with any system presented before. This is remarkable, especially considering that SLIDE achieves two-photon FLIM at up to 88 MHz pixel rate.

3) *"Following on from the previous concern, the authors have also failed to cite significant achievements in the literature from researchers working on high-speed multiphoton and lifetime imaging. [Giacomelli et al., Biomed Opt Express 6(11):4317, 2015], [Bowman et al., Nature Comms 10:4561, 2019], [Poland et al., Biomed Opt Express 6(2):277, 2015], [Krstajic et al., Opt Lett 40(18):4305 2015] and [Raspe et al., Nature Methods 13:501, 2016] would all seem to be relevant to this paper; I'm sure the authors can find more. A good literature review is at the heart of good research, and it is important for the wide readership of Nature Communications that this work is placed in its proper context."*

Author's response:

We agree that it is helpful to include citations to other important works and significant findings. We thank the reviewer to compose this short list of relevant papers. We included the citations. However, we want to elaborate on our initial stance on citations: since this is not a review article and citation count is limited, we aimed at citing only work that has influenced this work or that is necessary to underpin a certain statement or claim. However, we agree that mentioning a few important works of fast TPM and FLIM can be helpful for the multidisciplinary reader to further delve into this field. Unfortunately, this list can never be comprehensive. We want to thank the reviewer for taking the time to look up these relevant manuscripts and thereby improving the scientific quality of the paper. We included these references along with other fast techniques and review articles on the subject.

4) *The drive for high pixel throughput is laudable, but one of the approaches commonly used to achieve this is to increase the laser power to produce more fluorescence photons per pulse. Since this degrades the resolution of the system, it would be important to quantify the spatial resolution in X,Y and Z, by imaging some sub-diffraction-limited fluorescent beads for example.*

Author's response:

We thank the reviewer for this question. However, we want to distinguish two important points from this statement: 1) The resolution is always measured using a specific excitation power and sample. We did in fact perform these measurements with 200nm beads and confirmed diffraction-limited performance, as reported in line 68.

2) However, the other reviewer's statement refers to the insight that when applying higher laser power, even out of focus light can be excited which leads to a decreased resolution. This will practically be conducted with dim samples, not fluorescence beads that show high labeling concentration and thus high brightness. We have not performed a sample dependent analysis of signal brightness, required laser power and thus testing of diffraction-limited performance. This is an interesting point to analyze in the future, but requires a thorough and sample-dependent in depth analysis. Such an analysis was not the aim of this study. The diffraction-limited performance was here investigated to confirm that the spectrally-dispersed line-scanning approach yields high fidelity, diffraction-limited performance in non-linear imaging. However, what the reviewer suggests here would be a sample dependent analysis of resolution which requires an a priori knowledge of the sample size (i.e. ideally sub-diffraction limit) in all three dimensions. This is in practice unknown, so fluorescent beads are commonly used, which however show high staining and thus typically don't produce this problem. Perhaps additionally the pollen grains in the supplementary show very thin spikes which are sub- μm at their tips. These are very well resolved with the SLIDE setup, as can be seen in sup.fig. 5.

There are also more minor concerns which should be addressed but are not as critical. These are as follows:

- 5) *Line 27 is debatable - the I^2 dependence of multiphoton excitation only favours single-point scanning if the focus remains unsaturated; because saturation is comparatively easy to achieve with the $\sim 1\text{-}5\text{W}$ available from modern femtosecond lasers, optimal use of laser power actually favours scanning of multiple spots simultaneously (a technique known as Multifocal Multiphoton Microscopy) and in the extreme case, favours excitation of a whole field of view at once (also known as Temporal Focusing). If the user is trying to image deep into tissue then these parallel techniques are unfavourable, owing to crosstalk between the foci, but this also doesn't particularly support the authors' need to justify their rapid imaging, as the number of fluorescence photons captured when scanning deep into tissue is much lower than at the surface, and hence pixel dwell times are not limited by the speed of the scanning system.*

Author's response:

With this statement we wanted to draw to the attention that a usual scaling principle of linear one photon fluorescence imaging – widefield illumination – is not necessarily possible in TPM due to the quadratic scaling. We believe that this is an important information for the multi-disciplinary readership. However, we agree with the reviewer's statement that MMM or temporal focusing makes more efficient use of the available laser power to generate high throughput, by parallelizing the excitation. However, this parallelization comes at a cost. Typically the resolution is given by the excitation beam already. By parallelizing the excitation, the resolution is governed by the detection. As the reviewer states, this resolution is generally decreased in scattering media.

Regarding the second point of achieving sufficient photon numbers (also in deep imaging) vs. saturation, a faster scanning can help immensely by avoiding dark states and thus lead to increased signal levels. This discussion on the usefulness of fast scanning was already

discussed above and we just want to refer again to the works by Stefan Hell's group and Arthur Konnerth's group on observation of signal increase due to fast scanning. A further important point is that array detectors are typically slower than PMTs and have less detection area per pixel, so not as well suited for FLIM imaging (although fast array detectors exist, e.g. SPAD-array detectors). However, most FLIM systems with high temporal resolution and high spatial resolution rely on point scanning techniques, so with this statement we wanted to highlight that SLIDE does not sacrifice these advantages of traditional laser scanning microscopy (single pixel per time) and FLIM (single pulse per time).

- 6) *The methods section should contain a concise description of the steps required to replicate the authors' results. Currently there is a great deal of material that would be better placed in the main body of the text, or at the very minimum in Supplementary Information. Furthermore, all parts in the system should be described - the manufacturer and the model number are required for each part. This is particularly important for the galvo scanner, which is an integral part of the high-speed scanning system at the heart of this paper.*

Author's response:

Thank you for catching it, we shifted around a good deal of information, which is better placed in the main text.

We further included the make and model of the galvo mirror. Finally, we double-checked to have included all the information needed to rebuild everything.

- 7) *There has been no attempt to determine whether the FLIM results are accurate - at the bare minimum, comparisons using a sample with a known fluorescence lifetime should be performed. There is a sentence (line 73) to say that this has been done, but no evidence is provided. In particular, photomultiplier tubes can have saturation effects at high gain (recording one photon decreases the probability of recording one within the next few hundred picoseconds) which might be manifested as a bias in the lifetime measurement when directly recording the fluorescence decay.*

Author's response:

We are sorry if our previous statement regarding this important point was unclear. A reference fluorescence measurement was indeed performed and compared to a literature value and similar molar concentration of the same solvent. We included the following line in the manuscript:

"We confirmed the recorded lifetime values for Rhodamine 6G to literature values in previous works to confirm the validity of the analogue lifetime detection approach."

Regarding the insight of lifetime bias due to photodetector saturation, we thank the reviewer for this comment. We haven't observed a lifetime discrepancy when measuring high photon counts, but will conduct an analysis on this in the future. Measuring a lifetime at different incident powers and adjusted averaging should yield same lifetimes. In other words, such a measurement would show different lifetimes at different photon counts per time if photodetector saturation plays a role. We will conduct such a measurement in the future, once we also have access to a TCSPC system for in depth quantitative lifetime assessment (see response to reviewer 1 above). In this proof-of-principle paper we focussed on

qualitative FLIM imaging at high speed to show imaging FLIM flow cytometry at very high speeds and its potential for cell sorting based on 2P-FLIM.

- 8) *“The fact that the laser cannot be tuned easily over a wide range of wavelengths is a limitation for practical use, and this should be mentioned in the text. ~1060nm excitation only permits excitation of dyes that lie towards the red end of the spectrum, leaving a great many green and blue dyes unusable. Since many important proteins (GCaMP, ChR2, GFP etc.) are green, this is a strong limitation for practical use of the technique and should be mentioned.”*

Author’s response:

This is a very valuable point and we made sure this is clearly written in the paper. We therefore included the following sentence to the manuscript (line 193):

“Also, the SLIDE system presented here has an excitation wavelength around 1060nm, however future laser developments will target excitation wavelengths at 780nm for autofluorescence applications and 940nm for GCaMP-based imaging.”

- 9) *Line 118 - the fact that the beads are approximately the same size as cells does not make them a good proxy for cells. Fluorescent beads are much brighter than even the brightest cells, and it is unlikely that imaging a 15um bead will provide useful information on the applicability of the technique to biological imaging.*

Author’s response:

Apart from sample brightness there are a couple of other things that are important for a spectral scanning system to show - the field-of-view that is achieved by the spectral span, the spatial resolution it provides governed by the instantaneous linewidth of the individual pulse and the diffraction grating resolution, the line repeatability or spectral revisiting stability, jitter etc.. These are all demonstrated in these measurements based on beads that have a comparable size. Sample brightness is just one other factor that is of interest and yes, these fluorescent beads yield high signal levels, but also other samples or staining methods lead to high signals (e.g. algae autofluorescence, Rhodamin staining, Alexa antibody staining etc.). So our experience does not confirm the reviewers statement that there is a signal level gap between all cells and these beads – perhaps the reviewer is referring to one photon applications of non-confocal nature.

However, the imaging of the algae cells with high sub-cellular resolution helps identifying the lipid droplets in distribution and size and is therefore perhaps an even more suiting proof-of-principle application for the SLIDE microscope in imaging flow cytometry using FLIM.

- 10) *It would be good practice to release all code, data and supporting information in an open source repository. This would aid other researchers in reproducing this work.*

Author’s response:

We are happy to share the code, data and any other knowledge on the system. However, due to the complexity of the system and refined synchronization, data structure etc. assistance from our side is needed in order to make sensible use of the data. We included this statement in the manuscript: “Data availability: The data that support the findings of this study are available from the corresponding author upon reasonable request.”

11) Line 441 - 'Mouse' is misspelled. Furthermore, it would be good to include or reference a detailed protocol for the transfection and other animal procedures.

Author's response:

Corrected. Thanks for catching it. We further added more details on the mouse preparation protocol.

12) The authors should be prepared to justify their choice of single-exponential fitting for all their lifetime measurements. Many fluorescent proteins are better fitted with a multiexponential distribution, so it is important to be sure that a monoexponential model is truly the best choice.

Author's response:

We have a double-exponential and deconvolution option implemented in our analysis software. Usually, we confirmed mono-exponential behavior either by plotting on a logarithmic scale and looking for “kinks” in the slope of the linear graph or by fitting with a deconvolved mono-exponential and looking at the residue. Most samples showed sufficient fitting fidelity already with mono-exponential model curves, see e.g. Supp. Fig. 4 of the fluorescence decay of tdTomato expressed in mouse neurons. Therefore, in most cases mono-exponential fitting was sufficient and always double-checked in detail.

13) I conclude by reiterating my view that this is the kind of research which should be published in high-profile journals such as Nature Communications, but a significant amount of extra work is needed before it is ready, as well as a more realistic and context-aware assessment of the capabilities of the new system. I look forward to reviewing a more polished version of this manuscript.

Author's response:

We want to thank the reviewer again for this thorough review and the help in improving the quality of this research manuscript.

Reviewers' comments:

Reviewer #1 (Remarks to the Author):

The authors have addressed all of my concerns satisfactorily and I am happy to recommend this revised manuscript for publication.

Reviewer #2 (Remarks to the Author):

The authors have done a fine job in addressing the concerns about their manuscript, but a few still remain:

1) I must disagree with their response to point 1). The authors make a largely sound argument that by reducing rep-rate (compared to normal multiphoton microscopes) while maintaining average power the peak power can be maintained (which consequently maintains excitation efficiency), but then completely neglect the fact that their system doesn't do that - it has an effective rep rate of 88MHz (a figure which isn't affected by binning), and uses power levels somewhere around 10-100mW (line 215). Consequently the comparison with a conventional 100fs 80MHz system using 1-50mW of average power is a largely valid one, and a substantial reduction in fluorescence excitation efficiency would be expected. I would expect their added paragraph to be amended to reflect this.

2) I must also reiterate my concern that this work does not represent an 88MHz pixel rate for FLIM, and in particular I would stress that it is not acceptable to bin pixels together for data analysis while still claiming the 'raw' pixel rate. Taken to extremes, that would mean I could disperse an ultrafast pulse over a 2048 x 2048 separate 'pixels', integrate the signal onto a single photodiode (essentially performing 2048 x 2048 pixel binning), plot the resulting fluorescence decay and claim I had obtained a 4 megapixel FLIM image in a few picoseconds. To be explicit, in Figure 3, the authors needed 3x3 pixel binning to get enough photons for a fluorescence lifetime measurement - this would imply 88MHz/9, or 9.8MHz effective pixel rate. For Figure 4, 9x9 binning was used (1.1MHz effective pixel rate) and for figure 5, 5x5 binning was used (3.5MHz effective pixel rate). Consequently, none of these figures truly represents a 88MHz FLIM pixel rate.

If the authors really wish to demonstrate 88MHz FLIM pixel rate, they would be able to take an image at 88MHz of a sample with a known lifetime, fit each pixel independently and show that their lifetime measurements were consistent with the literature value for a large number of pixels (ideally all of them). Alternatively I would accept a clarifying statement that while it may be possible to obtain a multiphoton image at a 88MHz pixel rate, FLIM images could not be obtained at the same

pixel throughput (and the various claims of an 88MHz FLIM pixel rate to be amended throughout the manuscript).

Leading on from the disagreement over what is a fair description of FLIM pixel throughput, the PCO.FLIM camera can take 1008 x 1008 images at >20fps without binning, which would imply a pixel rate of >20MHz. It is therefore important to be clear as to exactly what this technique is capable of, as by the authors' definition SLIDE is faster than the PCO.FLIM, whereas by disallowing binning it is a factor of 5-10 slower.

3) While I think the authors have technically fulfilled my concern that the other papers aren't being cited (and I do agree that this is not a review paper), I do not think it unreasonable to have a short paragraph summarizing the state of the art, rather than summarizing a whole field of work in the phrase "however faster frame-rates are desirable 6-23". If the reviewers are inventing a new technique, it is important that readers from outside the field (which is the vast majority in the case of Nature Communications) be made aware of what else has been achieved, and the standard by which this work is being measured. I do admit that there is an unfortunate modern trend towards obscuring the work of others in order to increase the apparent impact of ones own work, but I am confident that the authors have enough original and notable results here to not have to stoop to this level.

4) I would like to answer this point in two phases. The first is that the PSF degradation is due to oversaturation of the fluorophores in the focal volume and is not sample brightness dependent. It is probably best argued as follows: consider a multiphoton microscope with an average power of X, which is exciting 10% of the fluorophores in the focal spot (note that it is a *fraction* of fluorophores excited, not an absolute number - if we halve the concentration of fluorophores, the signal drops by half). If we increase the power to 20X, do we see 200% of the fluorophores emitting light? Clearly this ludicrous, and the effect of focus saturation is well-known in the literature (Cianci et al, Microsc Res Tech. 2004 Jun 1;64(2):135-141)

With regards to the second phase of this answer, it is not acceptable in a scientific paper to take a measurement (such as of a PSF) and just claim that it supports a particular conclusion (i.e. it is diffraction limited) in the text (now line 105). There is ample space in the supplementary information for a simple figure showing this data; it should take very little time to produce (since the authors already have the data) and readers are more than capable of assessing the claims of whether the system is diffraction-limited for themselves.

5) While I disagree with the claim that the resolution of MMM is given by the resolution of the detector (which it most definitely is not) these are minor points and I am prepared to let them go.

6) I am largely satisfied with these changes, although the make and model of the lenses in the beam expander and the grating should really be included as well. Arguably Page 17 doesn't belong in the Methods section, but if the editor doesn't object, then I do not wish to expend energy in enforcing journal style standards.

7) Once again, I should reiterate that it is not acceptable to perform a measurement, state its conclusion and expect the reader to trust that the interpretation is accurate (and that the statistical analysis has been performed correctly). In a scientific journal all reasonable characterization data should be provided, especially in the modern era of digital distribution when supplementary information can be as long as desired. The authors clearly have the data; it should take minimal effort to present it.

8) I am willing to accept this response.

9) I stand by my point that fluorescent beads (which are stained to highly optimal concentrations with extremely bright dyes) are brighter than almost all cell samples. I will concede that I've never tried imaging algae, but even the brightness of rhodamine-6G phalloidin (for example) or antibody-labelled dyes (which label at a far-reduced concentration compared to the dye concentration in a bead) pales in comparison to fluorescent beads. Nevertheless, I would be willing to accept a simple statement that the authors are not trying to use the beads as a proxy for cells in terms of brightness, and call the matter closed.

10) I commend the authors on this responsible position.

11) Correction accepted.

12) This process (of checking monoexponential fits for strange residuals or kinks) should be noted in the Methods section, as it is part of their experimental protocol.

Response to Reviewer

First of all, we would like to thank the reviewer for the thorough examination of our manuscript and the deep analysis of the underlying physical and optical implications. The quality of the paper as well as the scientific value for the community has greatly profited from the reviewers time and effort. We appreciate all the comments and took utmost care to address all of them fully. Please find our answers one-by-one below.

Reviewer #2 (Remarks to the Author):

The authors have done a fine job in addressing the concerns about their manuscript, but a few still remain:

1) I must disagree with their response to point 1). The authors make a largely sound argument that by reducing rep-rate (compared to normal multiphoton microscopes) while maintaining average power the peak power can be maintained (which consequently maintains excitation efficiency), but then completely neglect the fact that their system doesn't do that - it has an effective rep rate of 88MHz (a figure which isn't affected by binning), and uses power levels somewhere around 10-100mW (line 215). Consequently the comparison with a conventional 100fs 80MHz system using 1-50mW of average power is a largely valid one, and a substantial reduction in fluorescence excitation efficiency would be expected. I would expect their added paragraph to be amended to reflect this.

Author's response:

We apologize if we were not clear in our statement. In the previous response to the first review we wanted to alleviate the reviewers concern that “*fluorescence scales inversely with pulse duration*”. The discussion in the response was thus solely aimed at the general argument about longer pulse lengths for TPM.

In order to address the reviewers' critique more clearly, we re-arranged and re-wrote the paragraph on the longer pulse lengths in the discussion. It now includes the clear sentence that “In SLIDE, the repetition rate is typically 88MHz and thus duty cycles are normally higher than in femtosecond TPM systems (i.e. lower peak powers in SLIDE).”

The whole paragraph now hopefully makes both points clear, which were raised by the reviewer, namely the suitability of longer pulse lengths for TPM and the higher average powers yet lower peak powers employed by SLIDE, or equivalently, the lower excitation rates or excitation efficiency. We hope the new paragraph now makes a convincing argument without neglecting any important aspects.

We thank the reviewer for raising these very important concerns. This will certainly increase the quality of the manuscript and the introduction of the SLIDE technique to the scientific community.

2) I must also reiterate my concern that this work does not represent an 88MHz pixel rate for FLIM, and in particular I would stress that it is not acceptable to bin pixels together for data analysis while still claiming the 'raw' pixel rate. Taken to extremes, that would mean I could disperse an ultrafast pulse over a 2048 x 2048 separate 'pixels', integrate the signal onto a single photodiode (essentially performing 2048 x 2048 pixel binning), plot the resulting fluorescence decay and claim I had obtained a 4 megapixel FLIM image in a few picoseconds. To be explicit, in Figure 3, the authors needed 3x3 pixel binning to get enough photons for a fluorescence lifetime measurement - this would imply 88MHz/9, or 9.8MHz effective pixel rate. For Figure 4, 9x9 binning was used (1.1MHz effective pixel rate) and for figure 5, 5x5 binning was used (3.5MHz effective pixel rate). Consequently, none of these figures truly represents a 88MHz FLIM pixel rate.

If the authors really wish to demonstrate 88MHz FLIM pixel rate, they would be able to take an image at 88MHz of a sample with a known lifetime, fit each pixel independently and show that their lifetime measurements were consistent with the literature value for a large number of pixels (ideally all of them). Alternatively I would accept a clarifying statement that while it may be possible to obtain a multiphoton image at a 88MHz pixel rate, FLIM images could not be obtained at the same pixel throughput (and the various claims of an 88MHz FLIM pixel rate to be amended throughout the manuscript).

Leading on from the disagreement over what is a fair description of FLIM pixel throughput, the PCO.FLIM camera can take 1008 x 1008 images at >20fps without binning, which would imply a pixel rate of >20MHz. It is therefore important to be clear as to exactly what this technique is capable of, as by the authors' definition SLIDE is faster than the PCO.FLIM, whereas by disallowing binning it is a factor of 5-10 slower.

Author's response:

The reviewer has a convincing argument and I agree that the scientific community has to stay vigilant to claims which could dilute known figures of merit, which here is the pixel rate. Perhaps the answer to the first review was focusing too much on the reviewers (somewhat exaggerated) critique that "it is simply not possible using this system to capture FLIM data at 88 million pixels per second". Indeed, it is technically possible, as we are already showing up to 10 photons per pulse at 88 MHz pulse rate. This number only needs to be increased by a factor of 10 to lead to FLIM imaging without pixel binning, which can experimentally be reached by linear, one photon imaging with SLIDE, thus also utilizing out of focus fluorophores. This can in the future be conducted with an FDML-MOPA in the visible, which we described recently (Karpf et al., Opt.Lett. 2019).

Here we apply pixel binning, so we concur in that we shouldn't make this pixel dwell time claim. We adjusted the claims throughout the manuscript. Just as a clarifying final remark on this subject, in the FLIM community pixel binning is a standard procedure and pixel rates, frame sizes and frame-rates are typically stated without accounting for the binning factor. I guess this has to do with the fact that the morphology or image resolution is typically given by the fluorescence image, where pixel numbers are real and counted without any binning. Only the color, derived from the fluorescence lifetime, is "washed out/blurred" by binning. So the image in FLIM is somewhat special in this regard and the usual figure of merit "pixel rate" is typically expressed by the pixel dwell time of the morphological, fluorescence

intensity image, since it is the practically more important metric than lifetime resolution. However, recent work suggests that this practice of sacrificing lifetime resolution may lead to sub-par biological interpretation in some cases (see Evers et al, ref. 5).

As fast FLIM imaging depends on many factors, as best summarized in the recent review article (ref. 14), we just want to highlight the technical capability of SLIDE for fast FLIM, namely the very fast scanning speed paired with the high detection speed. So we removed the claim of fastest FLIM and now we only claim that it: *“enables recording of second harmonic generation (SHG), Two-Photon fluorescence and fluorescent lifetime imaging (FLIM) at speeds up to the excitation rate of 88 million pixels per second.”*

Also, we added a clarifying statement as suggested by the reviewer (line 259):

“In fact, the high speed of 88MHz pixel rate was achieved for two-photon fluorescence imaging, yet for FLIM imaging pixel binning was applied to achieve the necessary photon numbers of >100 photons per pixel. This blurs the lifetime (colour) resolution, without compromising the morphological resolution, which originates from the fluorescence intensity images.”

We hope that the reviewer can agree to this, as his/her comparison to the PCO.FLIM camera also only stated the theoretical speed, not a practically achieved speed in FLIM imaging.

As a side note, we understand the *reductio ad absurdum* of the ultrafast pulse over 2048x2048 pixels and later binning with a single photodetector, but this neglects the fact that this wouldn't produce an image. In FLIM, the image information (morphology, resolution etc.) is generated without binning and here SLIDE as a laser scanning technique achieves remarkable speeds over millions of resolvable points per second while maintaining very high resolution.

3) While I think the authors have technically fulfilled my concern that the other papers aren't being cited (and I do agree that this is not a review paper), I do not think it unreasonable to have a short paragraph summarizing the state of the art, rather than summarizing a whole field of work in the phrase "however faster frame-rates are desirable 6-23". If the reviewers are inventing a new technique, it is important that readers from outside the field (which is the vast majority in the case of Nature Communications) be made aware of what else has been achieved, and the standard by which this work is being measured. I do admit that there is an unfortunate modern trend towards obscuring the work of others in order to increase the apparent impact of ones own work, but I am confident that the authors have enough original and notable results here to not have to stoop to this level.

Author's response:

We expended this part to a paragraph highlighting the main working principles behind the speed improvements of the various systems cited.

4) I would like to answer this point in two phases. The first is that the PSF degradation is due to oversaturation of the fluorophores in the focal volume and is not sample brightness dependent. It is probably best argued as follows: consider a multiphoton microscope with an

average power of X, which is exciting 10% of the fluorophores in the focal spot (note that it is a *fraction* of fluorophores excited, not an absolute number - if we halve the concentration of fluorophores, the signal drops by half). If we increase the power to 20X, do we see 200% of the fluorophores emitting light? Clearly this is ludicrous, and the effect of focus saturation is well-known in the literature (Cianci et al, *Microsc Res Tech.* 2004 Jun 1;64(2):135-141)

With regards to the second phase of this answer, it is not acceptable in a scientific paper to take a measurement (such as of a PSF) and just claim that it supports a particular conclusion (i.e. it is diffraction limited) in the text (now line 105). There is ample space in the supplementary information for a simple figure showing this data; it should take very little time to produce (since the authors already have the data) and readers are more than capable of assessing the claims of whether the system is diffraction-limited for themselves.

Author's response:

Thank you for the suggestion, we included the figure in the supplementary. This was also important as we spotted an inconsistency in the manuscript. In line 104 ff. "*The x-axis imaging resolution is given...*" we had already argued that the x-axis resolution is given by the spectral characteristics of the angular dispersion scanning, which limits the x-axis resolution to a value above the diffraction limit. We corrected the mistake and updated the mentioned resolutions throughout the manuscript.

5) While I disagree with the claim that the resolution of MMM is given by the resolution of the detector (which it most definitely is not) these are minor points and I am prepared to let them go.

6) I am largely satisfied with these changes, although the make and model of the lenses in the beam expander and the grating should really be included as well. Arguably Page 17 doesn't belong in the Methods section, but if the editor doesn't object, then I do not wish to expend energy in enforcing journal style standards.

Author's response:

We included the make and model of the lenses and the grating, thanks for spotting this. Regarding page 17: good idea, we moved it up to the main text.

7) Once again, I should reiterate that it is not acceptable to perform a measurement, state its conclusion and expect the reader to trust that the interpretation is accurate (and that the statistical analysis has been performed correctly). In a scientific journal all reasonable characterization data should be provided, especially in the modern era of digital distribution when supplementary information can be as long as desired. The authors clearly have the data; it should take minimal effort to present it.

Author's response:

We apologize if we were unclear in the previous correspondence. Indeed, we referenced to the data which is part of a previous paper of ours but perhaps the wording was unclear. We changed the sentence to make it clearer and point the interested reader to the data. It now

reads: "The validity of the analogue lifetime detection approach was already confirmed in a previous work, where we compared the recorded lifetime value for Rhodamine 6G to a literature value in good agreement."

8) I am willing to accept this response.

9) I stand by my point that fluorescent beads (which are stained to highly optimal concentrations with extremely bright dyes) are brighter than almost all cell samples. I will concede that I've never tried imaging algae, but even the brightness of rhodamine-6G phalloidin (for example) or antibody-labelled dyes (which label at a far-reduced concentration compared to the dye concentration in a bead) pales in comparison to fluorescent beads. Nevertheless, I would be willing to accept a simple statement that the authors are not trying to use the beads as a proxy for cells in terms of brightness, and call the matter closed.

Author's response:

We changed the original sentence according to the reviewers' suggestion. It now reads: "The beads range in size from 2 to 15 μ m and are thus chosen as examples for typical mammalian cell sizes, although perhaps not comparable in terms of brightness."

10) I commend the authors on this responsible position.

11) Correction accepted.

12) This process (of checking monoexponential fits for strange residuals or kinks) should be noted in the Methods section, as it is part of their experimental protocol.

Author's response:

We added a sentence to the "Data Processing" paragraph in the Methods section.

REVIEWERS' COMMENTS:

Reviewer #2 (Remarks to the Author):

In this revision, the authors have addressed the remaining concerns to my satisfaction. I congratulate them on their excellent paper, and am sure it will be of widespread interest.